# Recent inland large lake outbursts on the Tibetan Plateau: Processes, causes and mechanisms

Fenglin Xu[1,2][†], Yong Liu[3][†], Guoqing Zhang[1], Ping Zhao[4], R. Iestyn Woolway[5],
Yani Zhu[6], Jianting Ju[1], Tao Zhou[1,2], Xue Wang[1,2], Wenfeng Chen[1]

[1]State Key Laboratory of Tibetan Plateau Earth System Science, Environment and Resources (TPESER),
Institute of Tibetan Plateau Research, Chinese Academy of Sciences, Beijing, China
[2]University of Chinese Academy of Sciences, Beijing, China
[3]Department of Atmospheric Science, School of Environmental Studies, China University of Geosciences,
Wuhan, China
[4]State Key Laboratory of Severe Weather, Chinese Academy of Meteorological Sciences, Beijing, China
[5]School of Ocean Sciences, Bangor University, Menai Bridge, Anglesey, UK
[6]National Meteorological Information Center, China Meteorological Administration, Beijing, China

[†]These authors contributed equally to this work.
**Correspondence**: Guoqing Zhang (guoqing.zhang@itpcas.ac.cn)

**Abstract.** Lake outburst events have been mainly focused on small glacial lakes in the Himalaya, while the historical events from inland large lakes are few and have received less attention. Inland large lakes on the Tibetan Plateau are expanding rapidly, with recent signs of increasing outburst risk, highlighting the need to elucidate the processes, causes and mechanisms to mitigate future impacts. Here, a long-term satellite lake mapping shows that the number and surface area of lakes on the Tibetan Plateau exhibit an increased trend over the past 50 years, peaking in 2023. Two notable outburst events occurred during this period: Zonag Lake (~150 km$^2$ in 2023) on 15 September 2011 and Selin Co (~2,465 km$^2$ in 2023, the largest lake in Tibet) on 21 September 2023. The cascading outburst of Zonag Lake caused its area to shrink by ~124 km$^2$ (-45%), while the downstream Yanhu Lake expanded by ~163 km$^2$ (+347%). The Selin Co outburst resulted in a water mass loss of ~0.3 Gt, the downstream Bange Co experienced a water level rise of ~2.3 m and an area expansion of ~18%. Despite its large water storage capacity, Selin Co experienced less water loss due to the flat terrain at the breach and the slow flow (~1 m/s at the damaged road), with an average discharge of ~154 m$^3$/s. Even with the low discharge, the Selin Co flood breached the lowland road within ~10 hours. In contrast, the large breach and steep terrain at Zonag Lake facilitated a rapid discharge of a sustained volume of water, with an average discharge of ~2,238 m$^3$/s. Selin Co resulted in only a short period of drainage reorganization, in contrast to the permanent reorganization caused by Zonag Lake. The underlying mechanisms of the increased precipitation as the main trigger for the two outburst events prior to the occurrence are different. For Zonag Lake, thermodynamic effects, i.e. changes in the atmospheric moisture, are the most important, while for Selin Co, dynamical effects, i.e. the vertical moisture motion induced by the changes in atmospheric circulation, dominate the precipitation patterns. Large lake outbursts in the Inner Tibetan Plateau are expected to increase in the near future due to the warmer and wetter climate, and urgent policy planning is needed to mitigate the potential future lake-induced flood damage.

**1 Introduction**

The Tibetan Plateau (TP) has the highest and most widely distributed cluster of plateau lakes on Earth, with more than 1,400 lakes larger than 1 km$^2$ and a total area of ~50,000 km$^2$, accounting for more than half of China's lake surface area (Ma et al., 2010; Zhang et al., 2019a; 2019b). These plateau lakes serve as important sentinels of climate change and early warning signals of abrupt transitions in the Earth system (Lei et al., 2014; Zhou et al., 2021; Loewen, 2023; Chen et al., 2022). Between the 1970s and 1990s, a decreasing trend in lake area on the TP was observed, mainly due to decreasing precipitation. Subsequently, especially since the mid-1990s, a dramatic expansion of lake area has been detected, driven by increasing precipitation and a continuous increase in cryospheric meltwater (Zhang et al., 2017b; Song et al., 2013; Chen et al., 2022). The significant expansion of these lakes could threaten the fragile ecological environment of the TP, particularly by inundating grasslands, altering water resources, disrupting habitats, and impacting biodiversity (Xu et al., 2024).

However, the current glacial lake outburst floods (GLOFs) in the Himalaya have received considerable attention due to their extensive historical record and enormous downstream damage (Veh et al., 2020; Zheng et al., 2021). Outburst floods of large inland lakes (i.e., those far from the glaciers) in the TP have received less attention due to their relatively infrequent occurrence. Historically, the expansion of inland lakes has been cumulative, with few outbursts occurring due to thresholds or limits not being reached. However, in recent years, signs of lake outbursts have become increasingly apparent, accompanied by a continuous and dramatic increases in lake levels and outward expansion of lake boundaries (Lei et al., 2023; Liu et al., 2016).

On 15 September 2011, an outburst flood event occurred in the Zonag Lake located in the Hoh Xi region of the TP (Liu et al., 2016; 2019; Wang et al., 2022a; Lu et al., 2021; Lu et al., 2020a). This outburst flood event altered the hydrological connectivity between lakes, linking Zonag Lake, downstream Kusai Lake, Hedin Noel Lake and Yanhu Lake (tailwater lake) (Lu et al., 2021; Chen et al., 2021b) (Figure 1). More recently, Selin Co, the largest lake in Tibet and the second largest lake in the TP, also experienced an outburst event on 21 September 2023, causing significant environmental and societal impacts. However, the processes and mechanisms behind these events are not well understood. Overall, current studies lack a comprehensive assessment and detailed examination of existing inland lake outburst events, which is essential to improve our knowledge of lake outburst and facilitate the development of early warning systems.

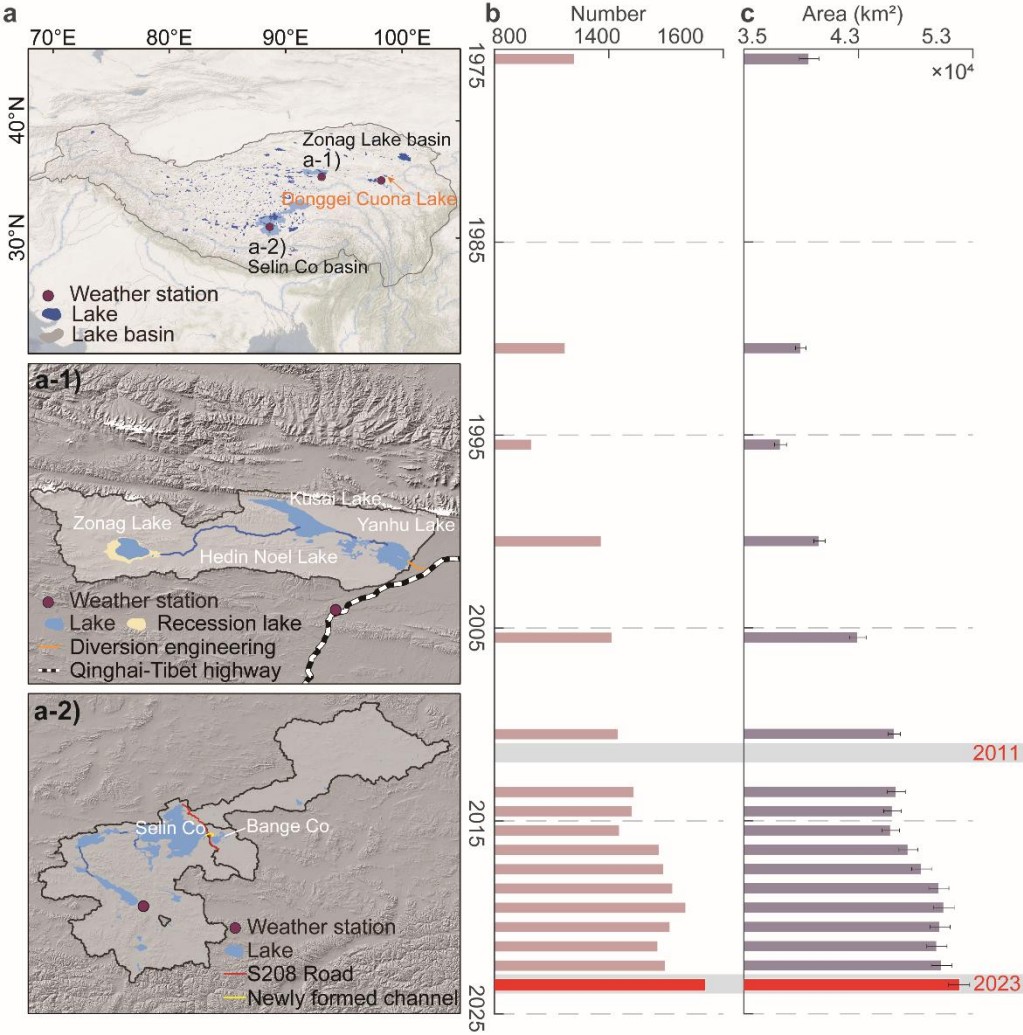

**Figure. 1**. Distribution and changes in number and area of lakes on the TP. (**a**) Location of the lakes on the TP, the Zonag Lake and Selin Co basin is shown in **a-1** and **a-2**. (**b**) Changes in the number of lakes (>1 km²). (**c**) Changes in area of lakes (>1 km²). Lake data from 1970 to 2018 are from Zhang et al. (2019a).

In a warmer and wetter world, with the associated increase in extreme climate events, lake outburst floods are expected to increase and pose a serious threat to the TP (Xu et al., 2024). A comprehensive understanding of the processes, causes, and mechanisms of such events is essential to improve our ability to predict and mitigate their hazard impacts. In this study, we focus on two notable inland lake outburst events: Zonag Lake on 15 September 2011 and Selin Co on 21 September 2023. The Donggei Cuona Lake event on 18 February 2024 is briefly discussed due to the human management of the lake and the relatively short duration of the outburst. The outburst processes are investigated by combining field surveys, remote sensing mapping and hydrodynamic modelling. The causes and mechanisms that triggered the two events are also examined, which can help to develop more accurate forecasting models that can better predict future extreme events by identifying specific conditions that precede outbursts.

## 2 Data and methods

### 2.1 Study area

Zonag Lake is situated in the north of Hoh Xil National Nature Reserve in Qinghai Province and the Sanjiangyuan National Park, where a large number of Tibetan antelopes are lambing, known as "Tibetan

antelope delivery room" (Chen et al., 2021b). On 15 September 2011, Zonag Lake burst, forming a connected hydrological system with Kusai Lake, Hedin Noel Lake and Yanhu Lake. Zonag-Yanhu Lake basin covers a total area of ~8,564 km$^2$, with an extensive distribution of permafrost and periglacial landforms.

The average elevation is 4600 m a.s.l. Based on nearby Wudaoliang meteorological station data from 1961 to 2019, the annual mean temperature is about -5.1℃ and the annual mean precipitation is 299.8 mm (Chen et al., 2021b). The region is characterized by three vegetation types: plateau meadows, grasslands and deserts (Liu et al., 2019).

Selin Co is the largest lake in Tibet and the second largest in the TP, with an area of ~2,465 km$^2$ in 2023.
On 21 September 2023, an outburst occurred at Selin Co, resulting in flooding downstream into eastern Bange Co. The total area of Selin Co basin is ~44,437 km$^2$, covering 369.7 km$^2$ of glacier and 13,404 km$^2$ of permafrost (Wang et al., 2022b). Selin Co is fed by four major rivers: Zhajiazangbu, Boquzangbu, Alizangbu and Zhagenzangbu, and has several upstream exorheic lakes, including Qiagui Lake, Wuru Lake and Cuoe Lake. The basin is dominated by a cold, semi-arid monsoon climate, with an average temperature of around
0 °C and average annual precipitation of ~350 mm (Tong et al., 2016). Donggei Cuona Lake is located in the western part of TP (98.55° E, 35.28° N), with an area of ~260 km$^2$ in 2023. Donggei Cuona Lake is an exorheic lake with a dam at the outlet. On 18 February 2024, a breach occurred in the left sluice chamber of Donggei Cuona Lake, causing water to flow out of the lake.

**2.2 Lake mapping from optical satellite data**

The lake area of the entire TP from the 1970s to 2023 was extracted using Landsat satellite images (Landsat 5/7/8/9, 30-m pixel), including 1970s, 1990, 1995, 2000, 2005, 2010, and 2013-2023, respectively (Zhang et al., 2019b). Compared to a previous study (covering 1970s-2018) (Zhang et al., 2019a), this dataset was extended to 2023, with more comprehensive records to reflect lake area changes in response to recent
climate change. Specifically, the lake mapping in the 1970s was derived from the Multispectral Scanner System sensors spanning 1972-1976. The lake extent from 1990 to 2010 with five-year intervals was mapped from Landsat Thematic Mapper images. With the launch of the Landsat 8 and 9 satellites, the available data quality has been significantly improved, allowing the annual lake outline to be mapped from 2013 to 2023. Based on the MODIS-observed cloud-cover cycle, it can be found that cloud cover is significantly higher in
summer (>50%) compared to autumn and winter (Zhang et al., 2017a). Additionally, lake outline extraction in spring and winter are often hampered by frozen ice cover and snow, resulting in inaccurate lake boundaries. Therefore, October images are prioritized due to the low cloud cover and relative stability of the lake area (annual maximum). If no images were available for October, images from the previous and subsequent months were also considered.

The Normalized Difference Water Index (NDWI) using the Near Infrared and Green bands was used to differentiate between water and non-water features, and the optimal thresholds for each lake were automatically determined using the OTSU method (Mcfeeters, 1996; Otsu, 1979). As lake boundaries are easily confused with mountains and cloud shadows, snow and ice, the lake outlines extracted in the previous step need to be visually checked and edited. Here, we manually examined each lake outline with reference
to the original Landsat image, and repaired incomplete lakes and removed to targets that were misidentified as lakes, to produce a high-quality lake dataset.

High-resolution PlanetScope satellite images with 3-m pixel and a 1-day cycle, were used to track the lake outburst process of Selin Co by visual interpretation, and to map the area change of Selin Co during 2022-2023 based on the NDWI index. Since June 2016, over 430+ Doves and SuperDoves sensors from
135 PlanetScope mission have been launched into 475-525 km sun-synchronous orbits, circling the earth every

90 minutes (Mullen et al., 2023). It provides four bands image including red, green, blue, and NIR channels, increasing to eight bands after 2020. Although PlanetScope has a high temporal resolution, it may not be able to completely cover the same region in one day. Therefore, the NDWI images within 1-11 days windows were synthesized into a scene of relatively complete NDWI images, and then a threshold of 0.02 (determined by comparing the accuracy of extraction boundaries based on the different thresholds) was used to produce lake boundaries in 2022-2023. In addition, images from Sentinel-2 (10-m pixel) (Yang et al., 2020) and Landsat-8 were also used to analyze the process of Donggei Cuona Lake outburst.

## 2.3 Lake level from satellite altimetry data

Sentinel-3 mission, comprising two identical polar orbiting satellites (Sentinel-3A and Sentinel-3B), was used to track changes in the water level of Bange Co and Donggei Cuona Lake. Sentinel-3A and Sentinel-3B were launched in February 2016 and April 2018, respectively, both equipped with a dual-frequency (Ku and C-band) Synthetic Aperture Radar Altimeter operating in open-loop mode with a cycle period of 27 days (Xu et al., 2021), providing high-quality observations of the lake water level. In this study, level-2 LAN_HY products at Non Time Critical timeliness were collected for the 2016-2024 periods. The water level of the lake with respect to the geoid (EGM2008) for the 20 Hz measurement was calculated using the following equation (1).

$$WL = H_{alt} - R - R_{geo} - N \qquad (1)$$

where *WL* is lake water level, $H_{alt}$ is the satellite altitude. *R* is the distance between the altimeter and the lake surface without corrections, $R_{geo}$ is the sum of geophysical and atmospheric corrections including the ionosphere, wet and dry troposphere, solid earth tide and pole tide. *N* is the EGM2008 geoid height referenced to the WGS84 ellipsoid. Subsequently, the 1.5 normalized median absolute deviation method was used to remove outliers and then calculate the along-track mean.

The area-water level linear relationship was used to obtain water levels from the 1970s to 2023 by inputting lake area (Zhang et al., 2021). The lake level of Selin Co is provided by Hydroweb (Crétaux et al., 2016; 2011).

## 2.4 Climate and auxiliary data

The daily in situ precipitation records from the China Meteorological Administration (https://www.cma.gov.cn) were used to the analyze long changes overs several decades and precipitation variations occurring shortly before the lake outburst event.. Meteorological data were obtained from the Shenzha station (30.95°N, 88.63°E) near Selin Co, Wudaoliang station (35.22°N, 93.08°E) near Zonag Lake, and Maduo station (34.92°N,98.22°E) near Donggei Cuona Lake (Figure 1). The data used covered the period from 1 January 1966 to 29 February 2024.

The monthly reanalysis dataset used in this study was derived from the fifth generation ECMWF atmospheric reanalysis (ERA5) (Hersbach et al., 2020), including geopotential heights, winds, special humidity, and other climate variables. The ERA5 data were employed in the diagnosis of climate mechanisms of the recent lake outburst events. The atmosphere was resolved using 137 levels from the surface up to a height of 80 km in the ERA5, with a grid of 31 km.

Gravity Recovery and Climate Experiment (GRACE) data was derived from Center for Space Research, University of Texas (CSR production) to estimate the change in terrestrial water storage (TWS) anomaly from 2003 to 2023 on the TP. CSR data have been widely used to estimate changes in water storage, drought and other fields because of their higher performance (Li et al., 2022; Pokhrel et al., 2021; Yi et al., 2016). In this study, the GRACE/GRACE-FO RL06.2 Mascon Solutions was used, which is an estimation solution expressed

in terms of spherical harmonic coefficients with a 0.25° grid. The Mascon solutions with all the appropriate corrections (GAD, GIA, C20, C30, degree1) were applied in equiangular grids.

**2.5 Lake outburst process from hydrodynamic model**

The Hydrological Engineering Center-River Analysis System (HEC-RAS), developed by the US Corps of Engineers (https://www.hec.usace.army.mil/software/hec-ras) was used to model the outburst process, including the flow path, depth and velocity. In HEC-RAS, an implicit finite volume algorithm was used to solve 2D Saint Venant equation to two-dimensional unsteady flow, which is well suited to the outburst processes. The dam failure module of HEC-RAS was used to simulate dam failure. Difficulties of HEC-RAS in modelling erosion processes on soils due to the high discharge of Zonag Lake, only the hydrodynamic process of the Selin Co discharge was simulated.

The model inputs include the lake storage, terrain, breach width and depth, breach formation time, manning roughness coefficient, and discharge hydrograph. For the Selin Co storage, only the change in water volume in the upper layer is required due to the considerable storage capacity and the relatively small water level drop (<0.5 m). This can be determined from the area, water level, and hypsometric curve. NASADEM was used as terrain data, as the NASADEM represents the topography in the year 2000 and the lakes on the TP experience significant expansion, the DEM values inside the interior of the lakes were replaced with elevations corresponding to the lake boundaries in 2023. The width and breach formation time of the breach was determined by tracking its progression using high-resolution PlanetScope imagery. Specifically, the breach width was approximately 60 m on 23 September 2024, expanded to ~160 m by 27 September, reached ~180 m by 3 October, and finally developed to ~200 m on 8 October. Based on this progression, we estimated that the breach formation time took 18 days beginning from the initial breach on 21 September and the width of breach was 200 m. The depth of the breach was set to 2 m, which is based on field measurements (Lei et al., 2024). The total calculation time was set from 21 September to 13 October 2023. The Manning roughness coefficient was set to 0.04 based on field surveys, which shows that the river channel area is covered by sparse herbaceous vegetation. As all the floodwater from Selin Co flows into Bange Co, the discharge hydrograph can be derived by calculating the increase in water volume in Bange Co after the outburst, which can be determined by area from Planetscope imagery and hypsometric curve. Due to the influence of clouds, shadow and other factors, only the average discharge over a limited period can be obtained.

To validate the reliability of the model, we implemented validation in three aspects. First, we compared the simulated water level changes in Bange Co with satellite-based measurements, as all floodwaters flowed into the lake. Second, the simulated flood path and extent were verified by comparing them with PlanetScope imagery to assess spatial consistency. Third, the simulated flood depth was validated with UAV-derived DEM measurements (the elevation differences between the riverbank and the riverbed can reflect the inundation depth of flood) of the post-breach flood channel.

**2.6 Climate diagnosis**

To elucidate the mechanistic drivers of the recent lake outburst events, the moisture budget diagnoses and two–dimensional Takaya–Namura were employed to investigate the dynamic and thermodynamic effects contributing to precipitation change (Takaya and Nakamura, 2001), as well as the changes in atmospheric circulation, respectively. The moisture budget equations are defined as follow:

$$P' = E' - \langle \boldsymbol{V}_h \cdot \nabla_h q \rangle' - \langle \omega \partial_p q \rangle' + \delta \qquad (2)$$

$$-\langle \boldsymbol{V}_h \cdot \nabla_h q \rangle' = -\langle \boldsymbol{V}_h' \cdot \nabla_h q \rangle - \langle \boldsymbol{V}_h \cdot \nabla_h q' \rangle - \langle \boldsymbol{V}_h' \cdot \nabla_h q' \rangle \qquad (3)$$

$$-\langle \omega \partial_p q \rangle' = -\langle \omega' \partial_p \bar{q} \rangle - \langle \bar{\omega} \partial_p q' \rangle - \langle \omega' \partial_p q' \rangle \tag{4}$$

where $P$ is precipitation, $E$ is evaporation, $V_h$, $q$, and $\omega$ denote horizontal winds, specific humidity, and vertical pressure velocity, respectively. The variables with a prime represent monthly anomalies. $\langle \rangle$ represents a vertical integration from the surface to 100 hPa. $-\langle V_h \cdot \nabla_h q \rangle'$ and $-\langle \omega \partial_p q \rangle'$ represent the changes in horizontal and vertical moisture advection, respectively. $\delta$ denotes the residual term. The changes in $-\langle \omega \partial_p q \rangle'$ can be divided into the thermodynamic ($-\langle \bar{\omega} \partial_p q' \rangle$), dynamic effects ($-\langle \omega' \partial_p \bar{q} \rangle$), and nonlinear component ($-\langle \omega' \partial_p q' \rangle$). The thermodynamic term reflects the contribution of moisture change, while the dynamic term represents the contribution of atmospheric circulation change.

The Takaya–Namura wave activity flux is defined as follows:

$$W = \frac{P}{2|U|} \begin{pmatrix} \bar{u}(\psi_x'^2 - \psi' \psi_{xx}') + \bar{v}(\psi_x' \psi_y' - \psi' \psi_{xy}') \\ \bar{u}(\psi_x' \psi_y' - \psi' \psi_{xy}') + \bar{v}(\psi_y'^2 - \psi' \psi_{yy}') \end{pmatrix} \tag{5}$$

where $U = (u, v)$ indicates the horizontal winds, and $\psi$ denotes the stream function. Overbars and primes represent the climatology means and monthly disturbances, respectively.

### 2.7 Field surveying

The field surveying was conducted in October 2023 and August 2024 for Selin Co outburst event. During the field surveying, we examined the breach, the flood channel, and damaged roads, and inlet of Bange Co. Uncrewed aerial vehicle (UAV) was used to capture the orthophotos of flood river channel from Selin Co to Bange Co, breach, as well as road and settlements at risk. Then, high accuracy DEM was generated from UAV orthophotos using Agisoft Metashape and Context Capture software to capture the depth and extent of Selin Co flood inundation.

### 3 Results

### 3.1 Recent remarkable responses of lakes to climate change

From the 1970s to 2023, lakes greater than 1 km$^2$ on the TP experienced a significant increase in their size. The TP also experienced the formation of many new lakes. The total number and area of lakes increased from 1080 to 1537 (+42%), and from 40,124±766 to 51,928±853 km$^2$ (+29%), respectively, during this time period (Figure 1b-c). The shrinking trend in the lake area was observed during 1970s-1995, followed by a continuous increase in 1995-2010. Thereafter, and despite inter-annual fluctuations, the overall trend in lake area continued to increase (Figure 2). These changes mainly occurred in the endorheic TP (a large, closed basin of ~700,000 km$^2$), which contains more than 73% of the number and area of lakes in the region (>1 km$^2$). It is important to emphasize that the expansion of lakes on the TP in 2023 is significant. The observed increase in the number and area of lakes was approximately 140 and 1,400 km$^2$, respectively, which exceeds the change observed within any given year during the past 50 years (Figures 1b-c, 2). Changes of lakes in the endorheic TP are the most pronounced. The lake changes also show a synchronous increase with the TWS from 2003 to 2023, especially in the endorheic TP (Figure S1).

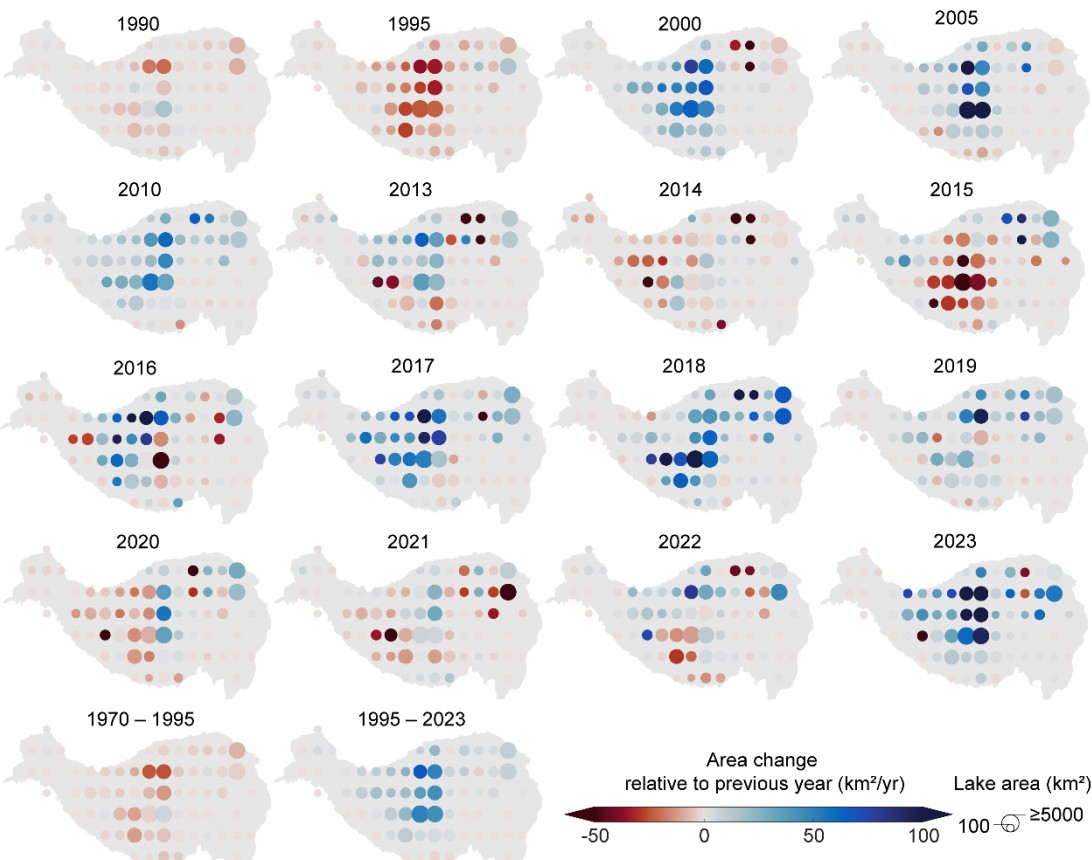

**Figure 2**. The spatial change of lake area on the TP. Lake area aggregated by 2×2° grid. Panels represent the average change rate in lake area relative to previous years with the available lake area data (km²/yr), divided by the time interval when a time span exists. For example, the first panel represents the changing ratio of area between 1990 and 1970. The last two panels represent the change in unit area from 1970-1995 and 1995-2023, respectively.

## 3.2 Process and cause of recent lake outbursts

With the rapid expansion of lakes on the TP, signs of lake outburst become increasingly apparent. These events attract widespread attention due to their impact and the surrounding environment. For example, the outburst of Zonag Lake on 15 September 2011, the outburst of Selin Co on 24 September 2023, and dam failure of Donggei Cuona Lake on 18 February 2024 have all been devastating. Here, we focus on the outburst events in Zonag Lake and Selin Co due to their wide-ranging impacts (i.e., damage to infrastructure, ecosystems, and biodiversity) and representativeness of the natural expansion of lakes on the TP.

### 3.2.1 Zonag Lake outburst event

From a long-term perspective, Zonag Lake remained relatively stable between 1975 and 2002 (with an average area of 257±1.55 km²), but then experienced an expanding trend, increasing to 274±1.42 km² on 22 August 2011 (Liu et al., 2016) (Figure 3a). On 15 September 2011, an outburst event occurred in Zonag Lake. A substantial amount of water from Zonag Lake was discharged in a short period of time due to the large width (~380 m) and depth (~27 m) of the breach and the steep gradient of the river channel (Figure S2) (Lu et al., 2021).The lake area of Zonag Lake reduced by ~107.52 km² observed by Chinese HJ-A/B satellites

satellite within 28 days (~3.84 km$^2$/day) (Liu et al., 2016), with water flowing downstream along the river channel into Kusai Lake. As Kusai Lake is recharged by floodwaters from Zonag Lake, it increased in size by ~38 km$^2$. The water then flows into Hedin Noel Lake in the adjacent basin by forming a new channel, which then flows into Yanhu Lake (tailwater Lake). This outburst directly altered the hydrological connectivity of the drainage system, merging two basins into one. Due to the continuous inflow of water into Yanhu Lake, its size increased from 47±1.26 km$^2$ in 2011 to 210±2.96 km$^2$ in 2019 (~20.37 km$^2$/yr).

The expansion of Zonag Lake was mainly attributed to the long-term increase in precipitation (Figure 3a), which led the lake to expand to its maximum extent, eventually causing it to overflow and burst. Extreme precipitation, defined as total annual precipitation above the 95th and 99th percentiles for the historical period from 1981 to 2010, also showed a significant increase in 2011 (Figure 4). This extreme precipitation played a key role in the lake's outburst. Additionally, two seismic events occurred prior to the breach: one on 27 July 2011 (62 km from Zonag Lake, magnitude 4.0), and another on 22 August 2011 (57 km from Zonag Lake, magnitude 3.1). These events may have weakened the geological stability of the lake dam (Liu et al., 2016). The contribution of glacier melt to Zonag Lake's expansion was relatively small. Model simulations indicate that glaciers and snow contribute around 21% to lake expansion (Wang et al., 2022a). Therefore, glacier melt plays a relatively minor role in the overall lake expansion process and outburst.

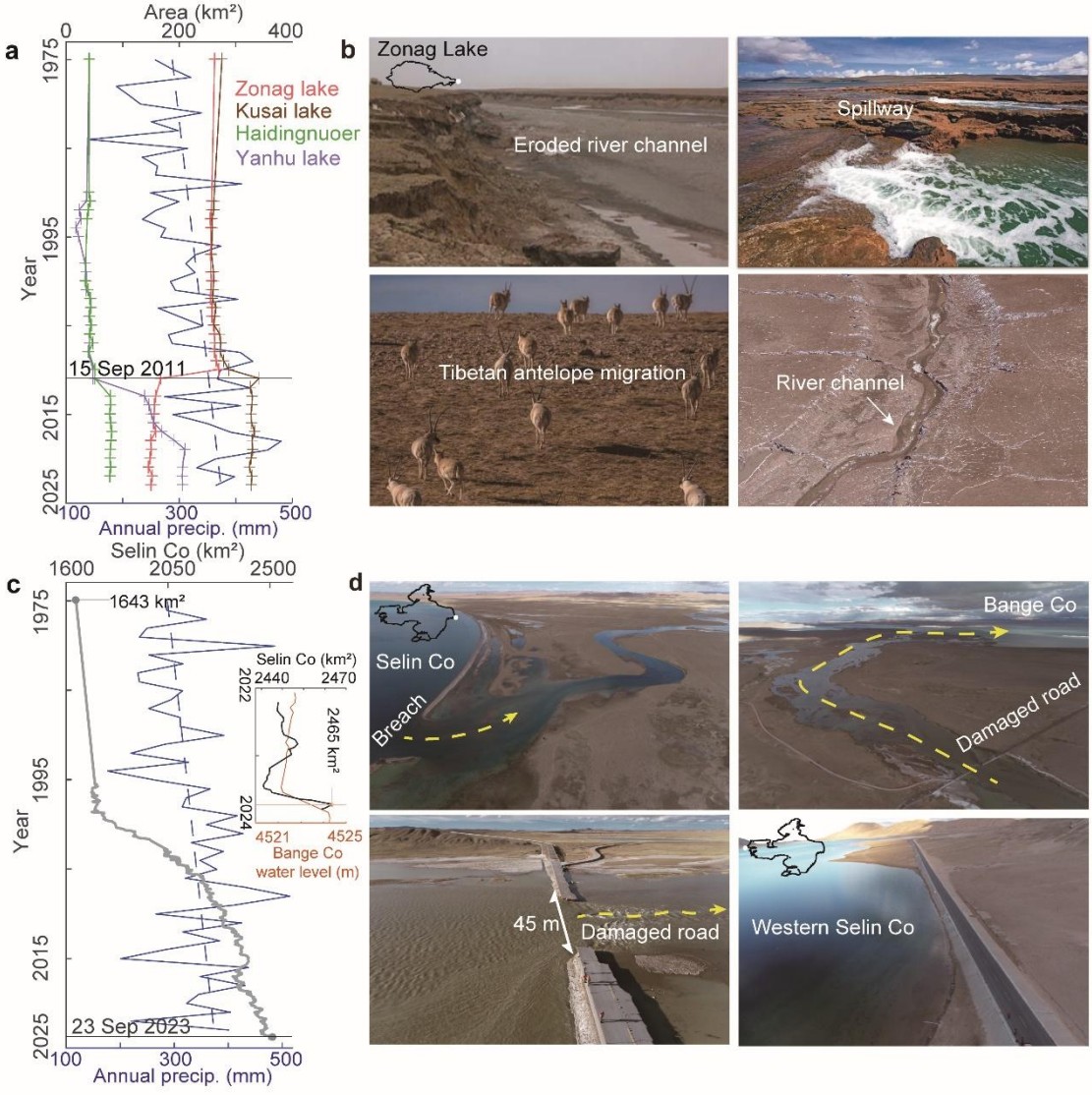

**Figure 3**. The changes and impacts of Zonag Lake and Selin Co. (**a**) The area change of Zonag Lake and downstream Kusai Lake, Haidingnuoer and Yanhu Lake, as well as the annual precipitation change from Wudaoliang weather stations (Blue). (**b**) The impact of Zonag Lake outburst, including the formation of new channel, soil erosion and the disruption of Tibetan antelope migration routes. (**c**) The area changes of Selin Co, as well as the annual precipitation change from Shenzha weather stations (Blue). The inset shows the rapid expansion of Selin Co and Bange Co in 2022-2024. (**d**) The impact of Selin Co outburst, including formed breach, damaged road, new channel between Selin Co and adjacent Bange Co. Selin Co has also approached near the road on the south-west side (lower right panel).

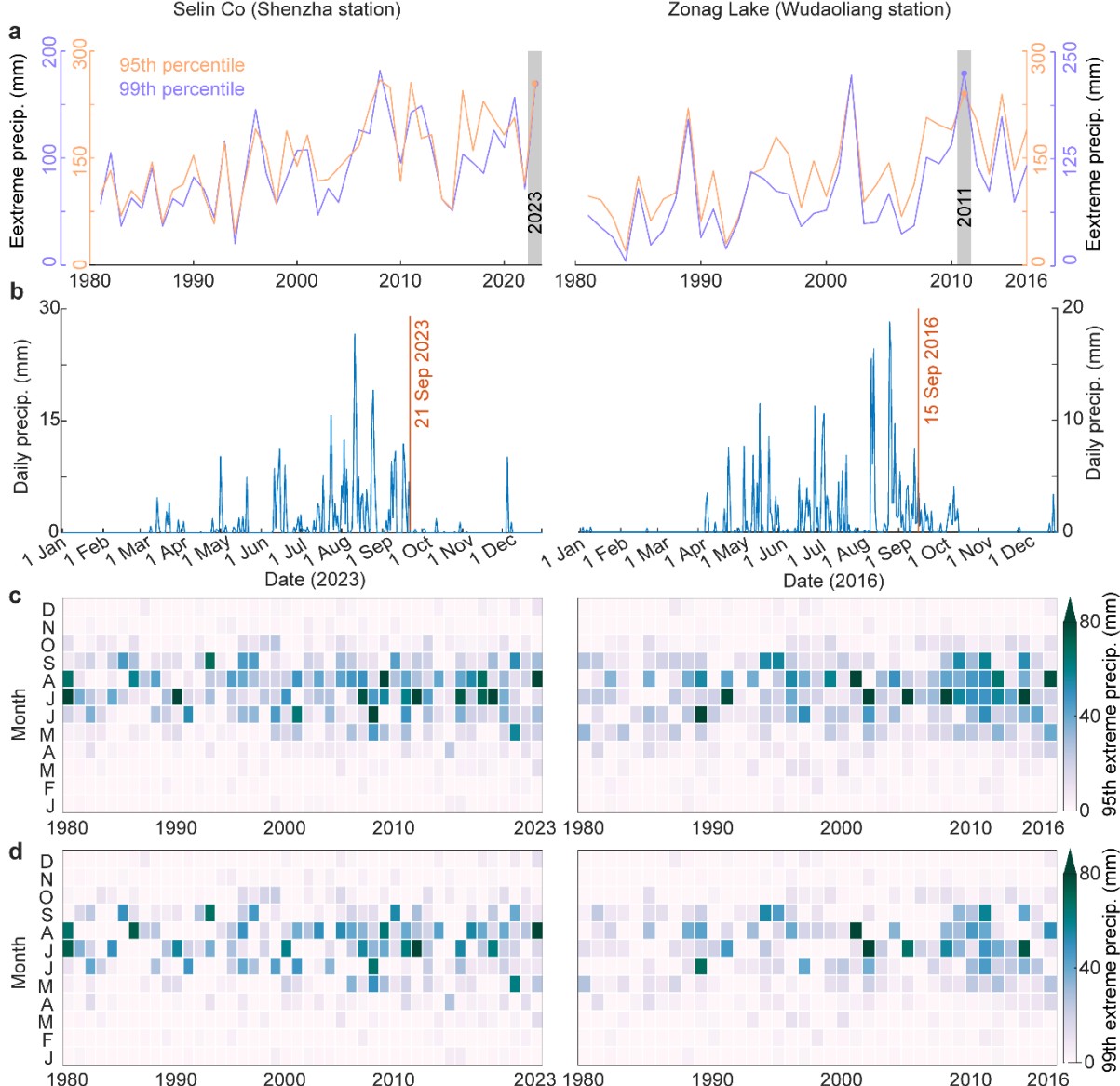

**Figure 4**. The change in extreme precipitation derived from Shenzha (near Selin Co) and Wudaoliang (near Zonag lake) weather stations. (**a**) Extreme precipitation (total precipitation that exceeds a 95th and 99th percentile during the historical period from 1981 to 2010) change from 1980 to 2023. (**b**) The daily change of precipitation prior to the outburst. (**c**) Monthly extreme precipitation change based on the 95th percentile from 1980 to 2023. (**d**) Monthly extreme precipitation change based on the 99th percentile from 1980 to 2023. The location of the weather station was shown in Figure 1.

### 3.2.2 Selin Co outburst event

Selin Co has experienced the most dramatic expansion (Figure 3b), with its area increasing by 50% from 1,643±4.52 km$^2$ in the 1970s to 2,465±10.82 km$^2$ in 2023 and water level rising by ~15 m. Due to both long-term sustained and short-term dramatic growth, the eastern side of Selin Co experienced an outburst event on 23 September 2023, resulting in the floodwaters breaking through the S208 road and flowing into the Bange Co. Based on high-resolution satellite images, this outburst process was identified (Figure 5). Selin Co was still in normal condition on 16 September, whereas by 20 September there was already a substantial amount of water near the S208 road indicating signs of heavy precipitation. By 21 September, Selin Co

experienced an outburst event. The breach width was found to be ~60 m by 23 September. The floodwaters initially overflowed the nearby road with continued pressure from lake flooding, the road was subsequently broken on 24 September, causing the floodwaters to directly flow into Bange Co. On 27 September the breach widened to ~160 m, and further expanded to ~200 m on 8 October. On 13 October, the flooding was stopped by the construction of an artificial dam (Figure 5g, h). This event resulted in a water mass loss of ~0.3 Gt in 23 days and a water level drop of ~0.12 m for Selin Co but led to a water level rise of ~2.3 m and an area expansion of ~18% for Bange Co (Figure 6).

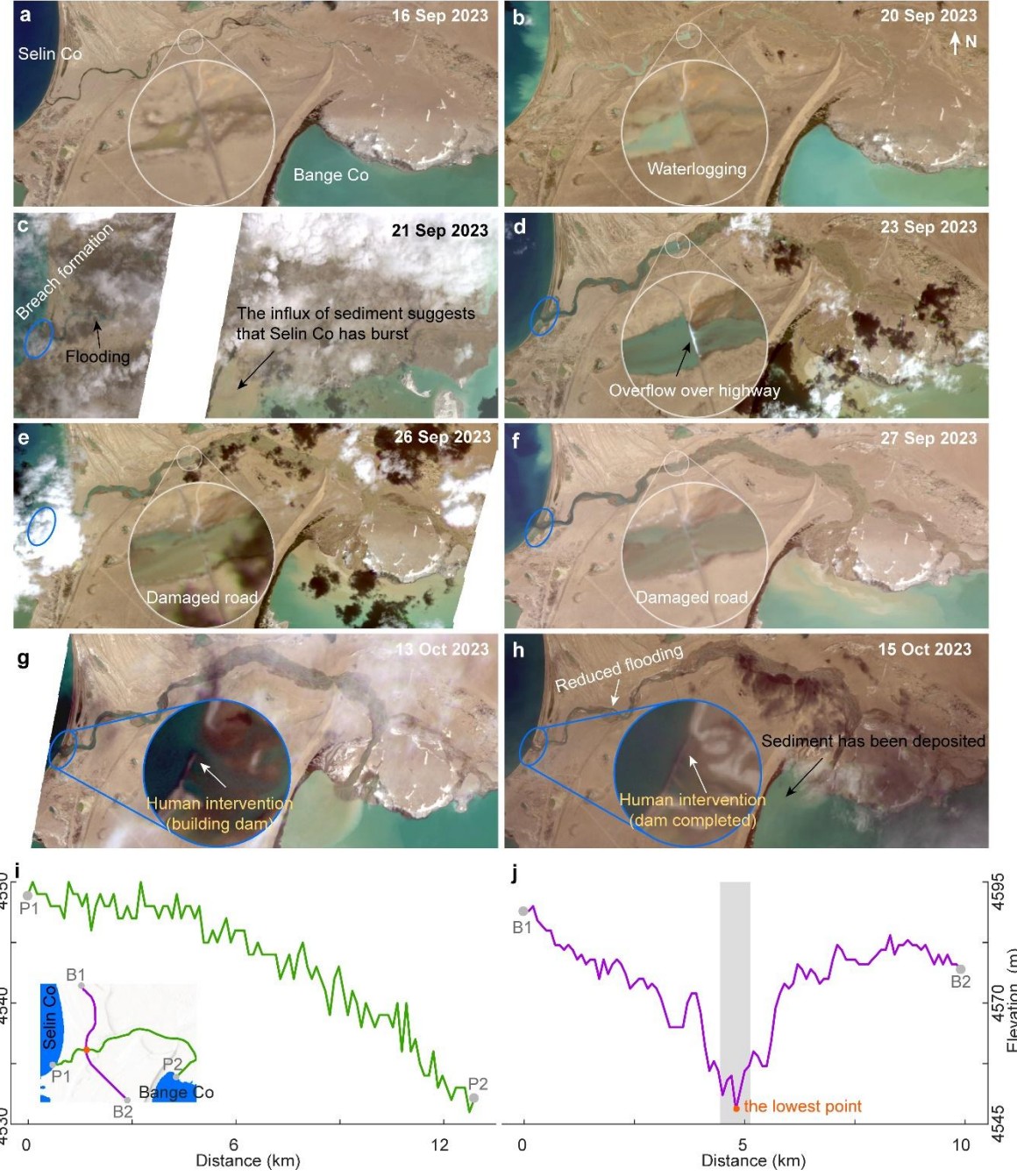

**Figure 5**. The outbrusting process of Selin Co as observed by PlantScope satellite imagery. (**a**) Selin Co remains normal. (**b**) Waterlogging due to precipitation along the roadside. (c) Sediment influx into Bange Co and flooding suggests that Selin Co has burst. (**d**) Flooding from Selin Co overflowing roads. (**e**) Flood from Selin Co destroys roads. (**f**) Floodwater from Selin Co continues to flow into Bange Co. (**g**) Construction of a dam

to block the floodwater. (**h**) Dams have been completed and flooding has been significantly reduced. The inset shows the damage process of the road caused by Selin Co or the newly built dam. (**i**) The elevation profile of the potential flow path between Selin Co and Bange Co before Selin Co burst. (**j**) The elevation profile of the common boundary between Selin Co and Bange basin. The inset in (**i**) shows the location of the profiles.

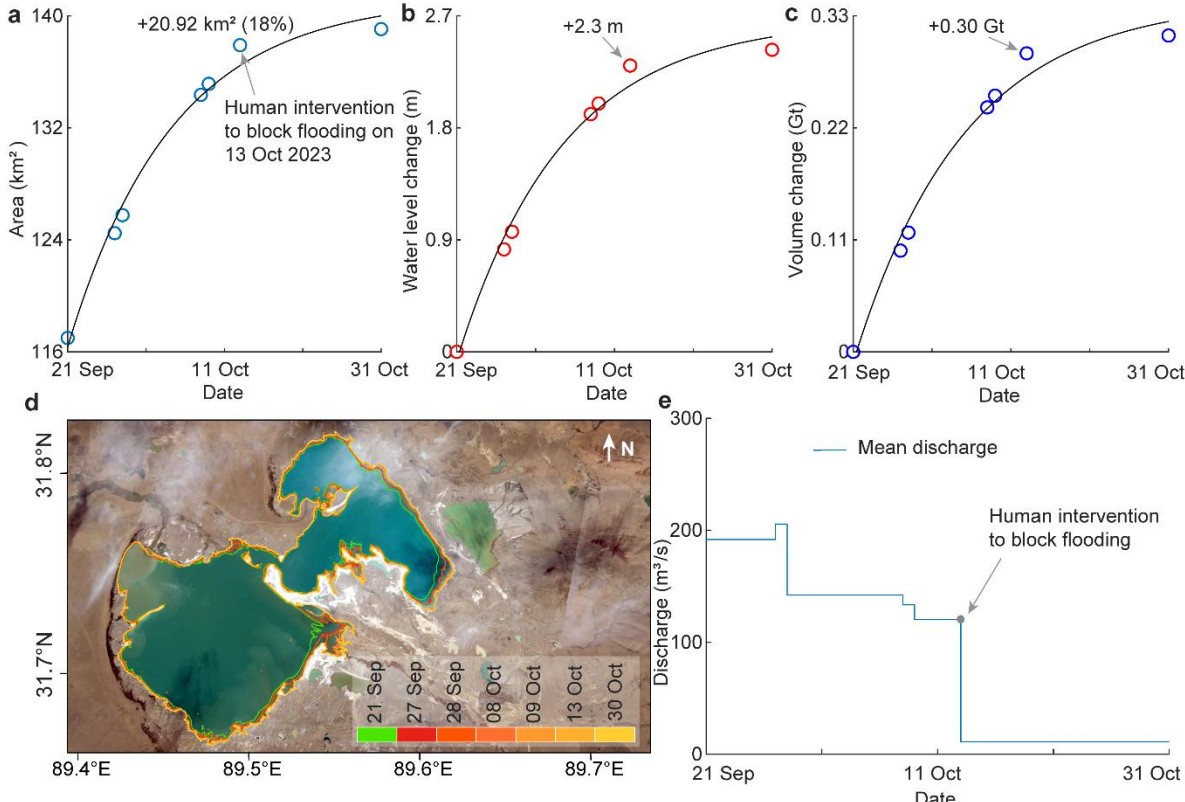

**Figure 6**. The water change of the downstream Bange Co after the Selin Co outburst and the discharge of the Selin Co flood. (**a**) The changes in area, (**b**) water level, (**c**) volume and (**d**) spatial extent of Bange Co after Selin Co outburst from the available PlanetScope images and hypsometric curve. (**e**) The mean discharge of the Selin Co outburst flood estimated from the volume change of Bange Co.

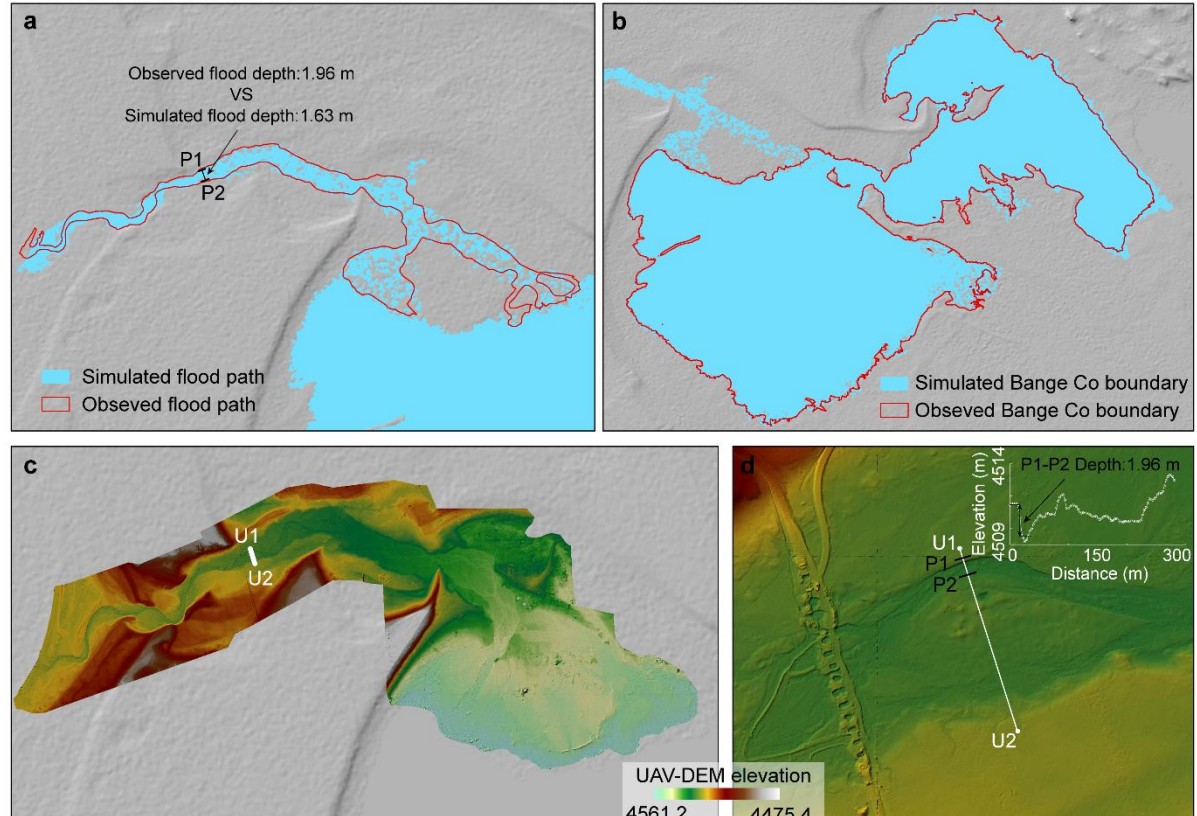

**Figure 7**. The validation of the Selin Co outburst flood simulation. (**a**) Comparison of the simulated flood path with PlanetScope observations of Selin Co on 13 October 2024. (**b**) Comparison of Bange Co boundary with PlanetScope observations of Selin Co on 13 October 2024. (**c**) The UAV-derived DEM of the Selin Co flood path. (**d**) The UAV-DEM near the damaged road, and the inset shows the elevation profile of the right side of the damaged road to indicate the flood inundation depth.

A water level rise of 2.23 m was identified from the hydrodynamic simulation, which matches well the observed increase of 2.30 m. Additionally, the simulated flood path and extent as well as the boundary of Bange Co showed good spatial consistency with PlanetScope imagery. We also compared the simulated flood depth with UAV-derived DEM measurements of the post-breach flood channel. The model simulated flood depth at the right side of the damaged road was 1.63 m, while UAV measurements indicated a depth of 1.96 m, showing overall good agreement (Figure 7). The maximum inundation depth at the damaged road reaches ~1.3 m, with a maximum flow velocity of ~1 m/s (Figure 8). Despite the modest velocity, the continuous pressure of the flood poses a significant threat, ultimately damaging the downstream road. The highest flow velocity occurs at the relatively steeply sloping inlet of Bange Co, reaching ~2 m/s. The simulation also shows that it takes 10 hours for the flood to reach the road and ~16 hours to reach the downstream Bange Co. This suggests that even in flat areas, the impact of flooding can be significant in a short period of time and requires the establishment of an early warning system. The outburst of Zonag Lake has permanently altered the basin reorganization. Flood waters are discharged in a short time at very high flow rates, making effective human intervention difficult. In contrast, the discharge of the Selin Co outburst was lower, and the Selin Co outburst caused only temporary basin reorganization or a brief transition from an endorheic to an exorheic lake due to human intervention. Both lakes were endorheic lakes and burst after expanding to be constrained by the terrain, emphasizing the need to monitor lakes with potential for

overflow. Overflow can exacerbate erosion at the breach, thereby increasing the risk of further breaches, particularly in lakes with steep breach gradients. Zonag Lake, with its large topographic gradient at the breach and large breach, had an extremely high discharge, averaging ~2,238 m³/s in 28 days. In contrast, Selin Co, despite its large storage capacity, had an average discharge of ~154.2550 m³/s in 23 days due to its flat terrain (Figure 6e). However, even at lower velocities, flooding could threaten roads within 10 hours, severely limiting emergency response.

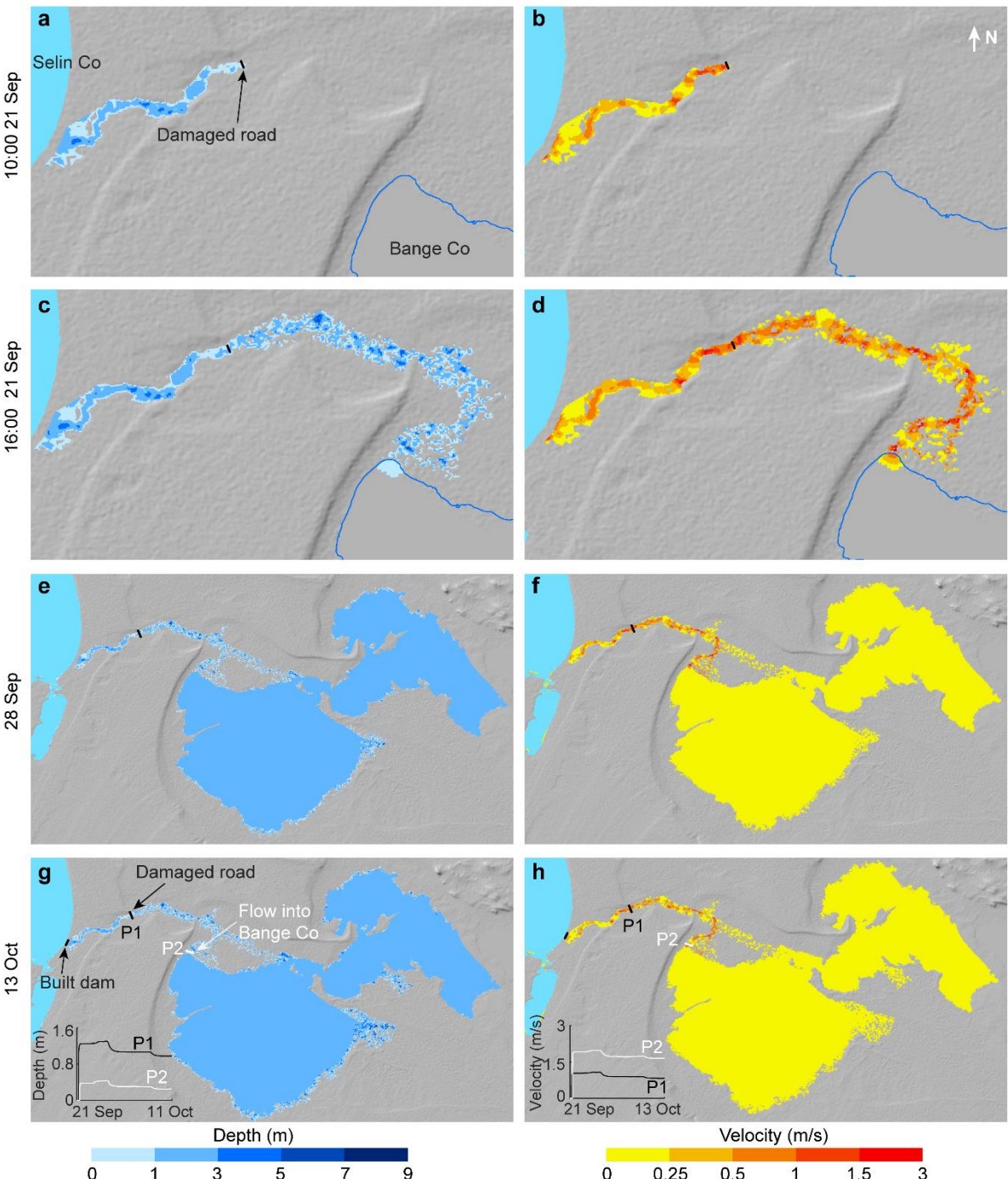

**Figure 8**. The outburst processes of the Selin Co outburst flood, including modelling of the discharge path, depth, and flow velocity. (**a, b**) The flood reaches the road. (**c, d**) The flood flows into the downstream Bange Co. (**e, f**) Flood simulation on 28 September 2023. (**g, h**) The flood was stopped by human intervention by

building a dam at the breach. The Inset shows the changes in depth and velocity near the damaged road and the mouth of the downstream Bange Co.

390 The expansion of Selin Co was similarly driven by a long-term increase in precipitation (Figure 3c), causing the lake to reach its maximum extent before overflowing and bursting (Figure 5i, j). Extreme precipitation increased as well, peaking in 2023 (Figure 4a), which also contributed to the lake's outburst. The contribution of glacier melt to Selin Co's expansion was relatively minor. DEMs derived from KH-9 and CoSSC-TanDEM-X data estimate glacier contributions to lake expansion at approximately 3.5% to 16.3%

395 (Chen et al., 2021a). Similarly, degree-day model simulations from 1976 to 2013 suggest a contribution of about 10% (Tong et al., 2016).

### 3.3 Climate mechanism of recent lake outburst

400 The Zonag Lake and Selin Co outbursts were triggered by the continuous expansion of the lakes due to a long-term increase in precipitation, as well as continuous heavy or extreme precipitation prior to the outbursts (Figure 4), underscores the necessity of investigating the climate mechanisms. Understanding the driving mechanisms of heavy precipitation is critical, not only to help explain the observed rainfall phenomena, but also to predict future flood risks under changing climate conditions. However, the atmospheric

405 mechanisms behind the increased precipitation during these two outburst events remain unclear. Here, we explore the contribution of each term of the moisture budget (equations 2-4) to the increased precipitation within Zonag lake basin in 2011 and Selin Co basin in 2023, respectively, thus providing a more comprehensive understanding of the factors leading to these outburst events.

 The increased precipitation observed in these two recent lake outburst events was primarily attributed to

410 the reinforced influence of vertical moisture advection (Figure 9). However, the contributions of thermodynamic ($-\langle\overline{\omega}\partial_p q'\rangle$) and dynamic effects ($-\langle\omega'\partial_p\overline{q}\rangle$) exhibited notable disparities between the two occurrences. The thermodynamic effect dominated the increased precipitation in Zonag Lake basin, highlighting the important role of the warming-induced increases in the atmospheric moisture content (Figures 9e, S4). In contrast, the dynamic effect contributed mostly to the increased precipitation in Selin Co

415 basin during 2023, indicating the prevailing influence of vertical motion resulted from changes in atmospheric circulation (Figures 9f, S5). It is noteworthy that horizontal moisture advection played a different role in the precipitation variations during the Zonag Lake and Selin Co outbursts, with a beneficial effect on the Zonag Lake event and a detrimental influence on the Selin Co event. This is evidenced by the anomalous moisture transport. In 2011, the enhanced southward moisture on the northern TP resulted in more moisture input in

420 Zonag Lake basin. In 2023, the prevalence of anomalous clockwise moisture flux circulation over the southern TP led to the northward transport of moisture towards the southern area of Selin Co, consequently triggering ascending motion. Simultaneously, the intensified westerlies over Selin Co gave rise to a net outflow of moisture, resulting in negative horizontal moisture advection.

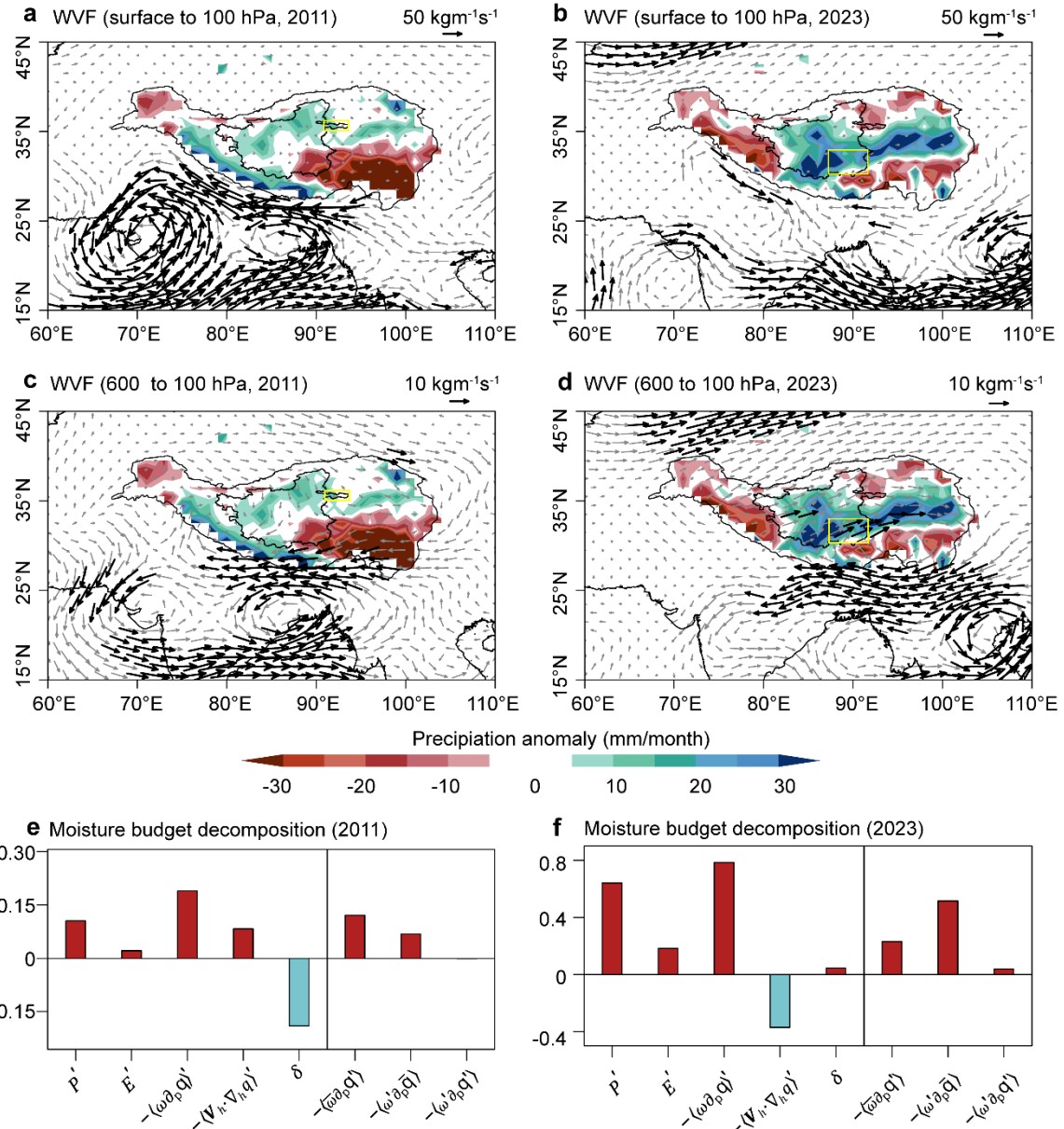

**Figure 9**. The atmospheric mechanism of precipitation-induced events in Zonag Lake and Selin Co. (**a, b, c, d**) Composite maps of anomalous vertically integrated moisture flux based on ERA5 data (WVF; integrated from surface to 100 hPa and from 600 to 100 hPa; vector; $kgm^{-1}s^{-1}$) in Zonag Lake basin during 2011 and Selin Co basin during 2023. The shading indicates precipitation anomalies. The black vectors indicate WVF exceeds the reference value. The reference climate state was selected as the average from 1981 to 2010. The yellow boxes represent the location of the Zonag Lake and Selin Co, respectively. (**e, f**) The contribution of each term (precipitation ($P'$), Evaporation ($E'$), the change in vertical ($-\langle \omega \partial_p q \rangle'$) and horizontal moisture advections ($-\langle \boldsymbol{V}_h \cdot \nabla_h q \rangle'$), residual term ($\delta$), thermodynamic $-\langle \bar{\omega} \, \partial_p q' \rangle$, dynamic ($-\langle \omega' \, \partial_p \bar{q} \rangle$), and the nonlinear effects ($-\langle \omega' \, \partial_p q' \rangle$)) of moisture budget components (mm/day) to precipitation changes averaged over the Zonag Lake and Selin Co basins.

Changes in regional atmospheric circulation play a pivotal role in modulating moisture transport and precipitation variations on the TP. Hence, we extended our inquiry to encompass the investigation of large-scale atmospheric circulation anomalies and the propagation of corresponding wave activity fluxes during the

Zonag Lake and Selin Co outbursts (Figures 10-11). A wave train emanating from the North Atlantic to Eurasia is evident in both 2011 and 2023; however, the propagation paths and strengths of these two wave trains exhibited notable disparities, with the wave train in 2011 being relatively flat and the wave train in 2023 being curved. In particular, the intensified wave activity fluxes over the East European Plain facilitated downstream wave propagation, leading to the establishment of negative potential geopotential height anomalies over Central Asia and positive potential geopotential height anomalies over the southern TP in 2023. This atmospheric circulation pattern was closely related to the moisture transport in the Selin Co basin, highlighting the important role of atmospheric circulation changes in lake outburst.

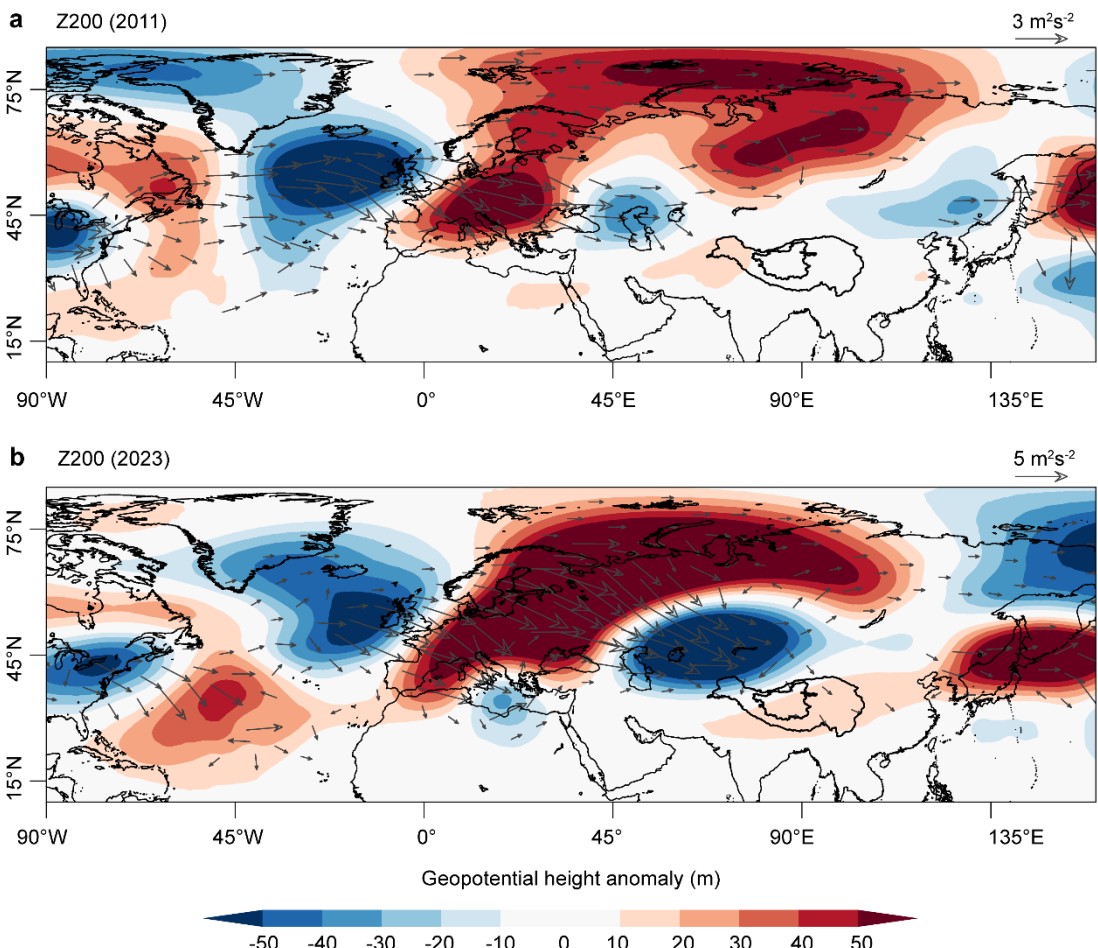

**Figure 10**. The spatial distribution of geopotential height anomaly (m) at 200 hPa in 2011 (**a**) and 2023 (**b**), resepctively. The arrows indicate wave activity fluxes. The reference climate state was selected as the average from 1981 to 2010.

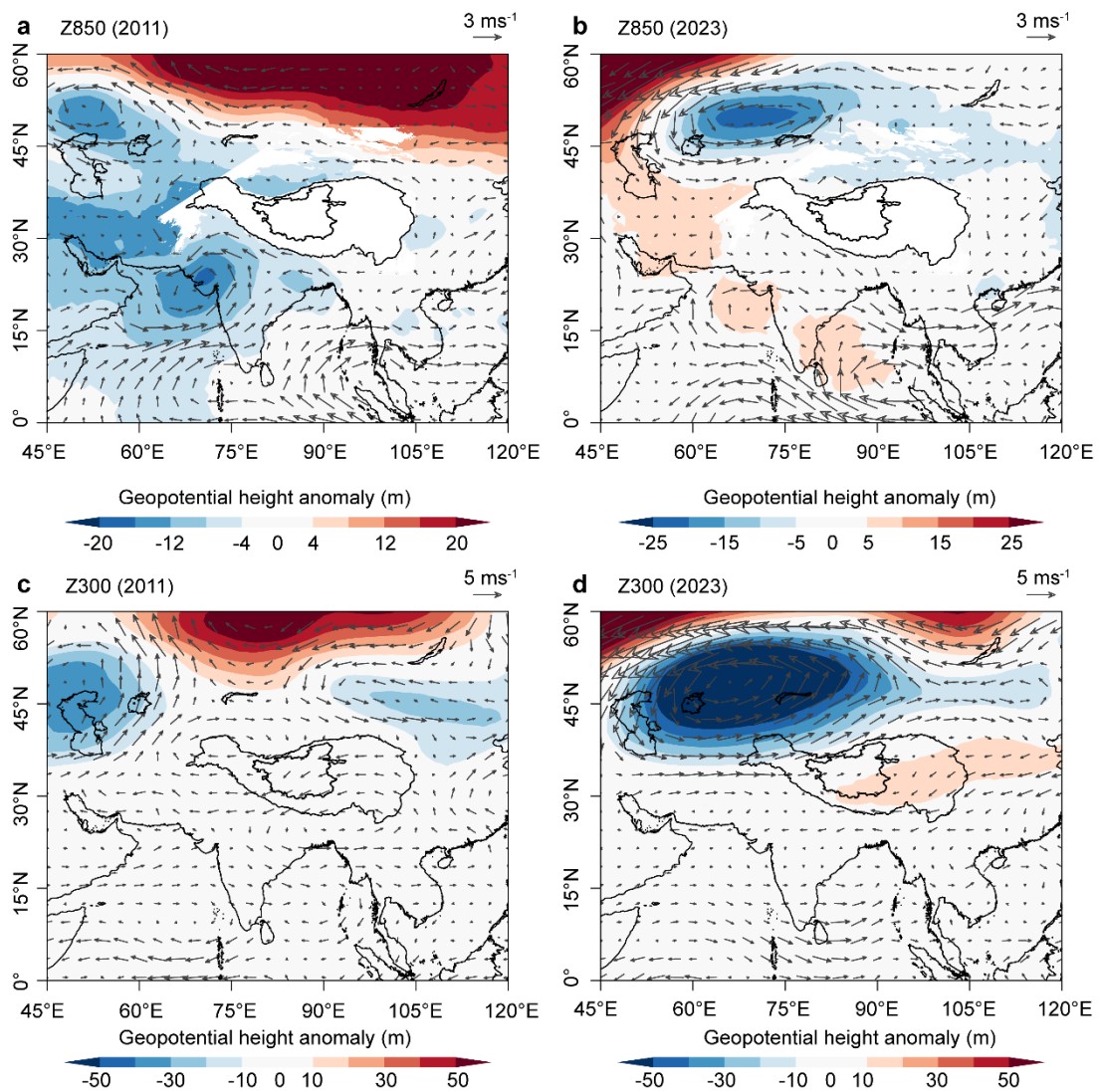

**Figure 11**. The spatial distribution of geopotential height anomaly (m) at 850 and 300 hPa, in 2011 and 2023, respectively. The arrows indicate wind vectors.

## 4 Discussion

### 4.1 Climate impact on recent lake outburst

Lake outbursts are mainly caused by the long-term, and sustained expansion of surface area driven primarily by increased precipitation, as well as short-term dramatic expansion due to extreme precipitation events. The decadal increase in precipitation on the TP can be attributed to the external forcing, Pacific Decadal Oscillation (PDO), and Atlantic Multidecadal Oscillation (AMO) (Liu et al., 2021; 2023). For example, the combined influence of external forcing and the PDO could result in an anomalous cyclone over the ITP and a weakened East Asian westerly jet, subsequently contributing to the decadal increase in precipitation in the Inner-TP (Liu et al., 2023). A positive phase of the AMO may lead to a northward shift and weakening of the subtropical westerly jet stream, which in turn affects moisture transport and results in changes in precipitation patterns (Liu et al., 2021; Sun et al., 2020). While the extreme precipitation events during the summer on the TP may be linked to the El Niño–Southern Oscillation (ENSO) (Figure S3) (Lei et al., 2019). The increased precipitation for the two events investigated in this study was largely attributed to enhanced vertical moisture advection, with distinct differences in the dominance of thermodynamic and dynamic effects.

Specifically, the thermodynamic effects played a more important role in the Zonag Lake outburst event, whereas dynamic effects predominated during the Selin Co outburst. It is evident that the increased precipitation resulting from climate warming and regional atmospheric circulation alternations has intensified lake expansion on the TP. Nonetheless, additional inquiry is warranted to reveal the distinct contributions of thermodynamic and dynamic effects to extreme lake expansion in various regions of the TP, particularly in understanding the mechanistic drivers of lake outburst events. This will be critical for understanding the response of the TP to climate warming.

**4.2 Consequences of Zonag Lake and Selin Co outburst**

Zonag Lake outburst has resulted in a cascade of consequences (Figure 3a). The ongoing expansion of the Yanhu Lake directly threatens the safety of the nearby Qinghai-Tibet Railway and highway (Lu et al., 2021). As a result, an artificial diversion channel was constructed for the Yanhu Lake in 2019 to channel water into the Yangtze Basin. Furthermore, this event accelerated the degradation of the permafrost surrounding the Yanhu Lake (Lu et al., 2020a) and facilitated the formation of new permafrost on the exposed lakebed of Zonag Lake (Zhang et al., 2022). The shrinkage of Zonag Lake directly caused sandstorms, exacerbating desertification (Lu et al., 2020b). Additionally, the connectivity between Zonag Lake and Kusai Lake, along with the erosion of river channels, has hindered the migration and reproduction of the Tibetan antelope (Liu et al., 2016). The outburst volume of 5.42 km³ from the Zonag event is exceptional in both a regional and a global context. According to a global inventory of GLOFs (Lützow et al., 2023), the largest recorded GLOF occurred in Iceland in 1726, with an estimated volume of ~25 km³, making it the only known event to exceed 5 km³. The outburst of an ice-dammed lake at Russell Fiord, North America, in 1986, with a similar water storage of ~5.4 km³ before breaching, released water with a peak discharge of 105,000 m³/s in one hour (Mayo, 1989). In the Tibetan Plateau, documented outburst volumes are much smaller, with the largest event mainly in the Karakoram, but only ~0.3 km³ (Lützow et al., 2023).

The Floodwaters of Selin Co flowed into Bango through the newly formed channel, merging the two basins into one and temporarily changing Selin Co from an endorheic to an exorheic lake (Figure 5). In addition, the Selin Co flood eroded the soil and carried substantial sediment into Bange Co, muddying its waters. Bange Co contains large quantities of high grade LiCl (lithium chloride), well above industry standards, and has significant development potential (Li et al., 2024). However, the continued inflow of Selin Co floodwater had diluted the concentration of LiCl in Bange Co, increasing production costs and reducing economic value. The damaged section of the road is located south of an intersection, disrupting travel on both two roads, and severely impacting the daily lives of residents. Local traffic was restored on 25 October following emergency repairs by government authorities by sealing the breach and building temporary roads. In addition, water in Selin Co reached the road on its southeastern side in October 2023 and is threatening to flood it (Figure 3b).

**4.3 Sign and recommendation for recent increasing lake outburst**

On 18 February 2024, the Donggei Cuona Lake in Qinghai Province experienced a burst in the floodgate on the left side of its dam due to expansion (Figure 12). Donggei Cuona Lake is an artificially influenced exorheic lake, with a dam constructed at its eastern outlet. It is important to emphasis that this outburst event occurred in winter when the lake surface was fully frozen. The area of Donggei Cuona Lake has increased by 14% from $229 \pm 1.40$ km$^2$ in 2000 to $260 \pm 3.33$ km$^2$ in 2023, and the water level has risen by 1.75 m, which is mainly attributed to the significant increase in precipitation. The nearest weather station to the Donggei Cuona Lake indicates that precipitation in 2023 reached 572 mm, setting the highest record in

the past 60 years, compared to a historical average of 340 mm. This outburst resulted in a significant increase in discharge pressure in the downstream river channels, which had substantial impacts on human living environments, affecting five townships and resulting in the death of approximately 195 livestock. Additionally, about 24.95 km of roads and pastoral paths and some water management facilities were destroyed by flooding, and some pastures were also inundated. Emergency repairs were promptly undertaken, and the damaged gate was sealed on 21 February (Wang, 2024).

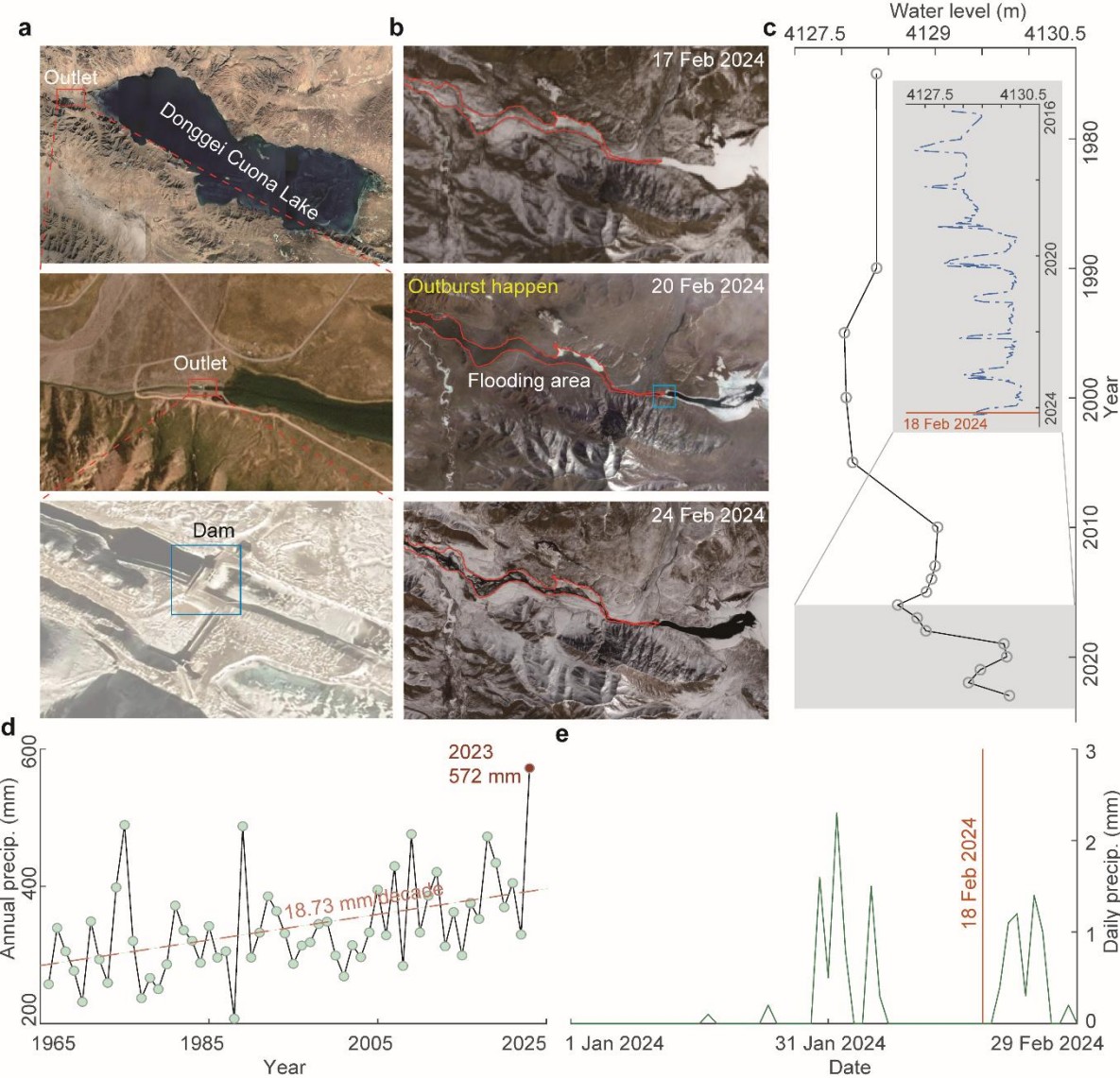

**Figure 12**. The location, outbrust process, and area change of Donggei Cuona lake. (**a**) The location, outlets, and dams of lakes. (**b**) The outbrust process based on the Landsat imagery on 17 February 2024, PlanetScope on 20 February 2024 and Sentinel-2 on 24 February 2024. (**c**) The change in lake water level from 1975 to 2023. The inset shows the water level change derived from Sentinel-3 data. (**d**) The change of annual precipitation from the Maduo weather station near Donggei Cuona Lake. (**e**) The daily precipitation from 1 January to 29 February 2024.

The outburst of Zonag Lake and Selin Co occurred in September, at the end of the monsoon season, indicating that lakes are at a higher risk of outburst following the expansion of lakes due to continuous rainfall during the monsoon period. In contrast, the outburst of Donggei Cuona Lake in winter exhibited distinct

characteristics from the monsoon season. The occurrence of this event while the lake surface remained entirely frozen suggests other factors facilitating the outburst, including vulnerabilities in the dam structure, sustained pressure on the dam due to rising water levels, and groundwater recharge in winter (Lei et al., 2022). The lake outburst outside the monsoon period may yield unexpected impacts on the surrounding environment and infrastructure, necessitating further in-depth research and monitoring to mitigate potential risks.

Lake outburst events on the TP can be divided into two categories based on their driving mechanisms: endorheic lake outbursts and GLOFs. The two documented endorheic lake outburst events in this region, Zonag Lake and Selin Co, both involved lakes expanding due to increased water input until they overflowed. This overflow led to erosion at the breach site, eventually resulting in full outbursts. For instance, in the Selin Co event, the breach widened from ~ 60 m on 23 September 2024 to ~200 m on 8 October 2024 due to ongoing erosion, which illustrates how overflow can rapidly escalate into a breach. In contrast, GLOFs are typically triggered by external factors such as ice avalanches and landslides. These events cause rapid destabilization of glacial lakes, leading to sudden, high-discharge floods that peak within hours or cause the end of flood releases. GLOFs tend to be more frequent due to the unstable nature of glaciers and other factors. For example, the Gongbatongsha Co outburst events in 2016 in China demonstrate the devastating potential of these sudden releases (Wang et al., 2024), which are often difficult to predict and have higher peak discharge compared to endorheic lake outbursts.

While both types of events pose significant risks, endorheic lake outbursts are more gradual in nature, driven by the internal dynamics of lake expansion. In contrast, GLOFs are driven by external forces, often resulting in more abrupt and catastrophic flooding. Understanding these differences is critical for risk assessment and management on the TP, as the two types of lakes require different monitoring and mitigation approaches. For instance, monitoring glacier dynamics and potential ice avalanches is crucial for GLOFs, whereas tracking lake water levels and identifying potential overflow points is key for managing endorheic lake outbursts. The lakes on the TP are expected to continue to expand in the future (Yang et al., 2018; Liu and Chen, 2022), which will further increase the challenge of lake outburst mitigation. The widespread or synchronous nature of lake expansion on the TP means that there is enormous pressure to mitigate lake outbursts (Zhang et al., 2017b). Therefore, it is necessary to strengthen the risk assessment of existing lakes and develop more effective disaster prevention and mitigation strategies, including improvement of monitoring and early warning systems, the reinforcement of lake embankments, and enhance of the emergency response capacity. At the same time, it is also necessary to conduct in-depth studies on the relationship between lake expansion and outburst events and strengthen prediction studies to better protect the ecological environment and the safety of people's lives and property on the TP and to promote sustainable development.

**5 Conclusions**

The TP is home to the highest and most numerous groups of high-attitude lakes on Earth, which have experienced significant expansion. From the 1970s to 2023, the number and area of lakes larger than 1 km$^2$ increased by ~42% and ~29%, respectively. In particular, the number and area of lakes in 2023 have reached new levels for the last 50 years, with 1,537 lakes larger than 1 km$^2$ and a total area of 51,928$\pm$853 km$^2$, showing an extreme expansion. The continued expansion of the lakes has led to increasing signs of outbursts, from Zonag Lake on 15 September 2011 to Selin Co on 21 September 2023 and Donggei Cuona Lake on 18 February 2024.

The Zonag Lake outburst resulted in a significant area reduction of ~45%, while the Yanhu Lake expanded dramatically by ~347% from 2011 to 2023. The Selin Co outburst caused a 2.3 m rise in the water

level and an 18% increase in the area of downstream Bange Co within 23 days, while the water level of Selin Co decreased by only ~0.12 m. Both Zonag Lake and Selin Co are endorheic lakes that expanded to the maximum capacity allowed by their surrounding terrain before the outburst. However, although the area of Zonag Lake is relatively small compared to Selin Co, the steep terrain and large fracture facilitated a rapid average discharge of ~2,238 m³/s, whereas the Selin Co outburst discharged more slowly at ~154 m³/s due to the flat topography of the fracture. Selin Co caused only a short period of basin reorganization due to human intervention, in contrast to Zonag Lake, which caused a permanent reorganization.

Lake outbursts are mainly due to a long-term increase in precipitation, while an increase in extreme precipitation events also accelerates the early onset of outbursts. The increase in precipitation is mainly attributed to enhanced vertical moisture advection. Thermodynamic effects dominated Zonag Lake, highlighting the important role of climate-induced increases in water content. In contrast, dynamic effects were the primary cause of increased precipitation in the Selin Co basin, indicating the dominance of vertical motions driven by changes in atmospheric circulation. Despite anthropogenic intervention in Donggei Cuona Lake, the rising water level due to increased precipitation also had a significant impact on the breach.

These lake outbursts had a cascade of consequences, such as the destruction of the S208 road by the flooding of Selin Co, the rapid expansion of the tailwater lake, the flooding of inundation of pastures, the death of livestock, the accelerated permafrost degradation, and the reorganization of the drainage system. The recent increase in lake outburst events suggests that lake outbursts on the TP may become more frequent and widespread in the future as lakes continue to expand. Therefore, prediction of future lake changes and potential outbursts are urgently needed.

**Code availability.** The codes for this study are available from XXX.

**Data Availability.** The lake number and area between the 1970s and 2020 produced in this study are available at https://doi.org/10.6084/m9.figshare.25939831. The original Landsat 5/7/8/9 images were downloaded from https://earthexplorer.usgs.gov. The PlanetScope satellite images used in this study were accessed through the Education and Research Program (https://www.planet.com/markets/education-and-research/#apply-now) and downloaded from https://www.planet.com/explorer. The Sentinel-2 images and Sentinel 3 SRAL altimetry data were obtained from https://browser.dataspace.copernicus.eu. ERA5 reanalysis dataset were downloaded from https://cds.climate.copernicus.eu/cdsapp#!/dataset/reanalysis-era5-pressure-levels-monthly-means. The water level and area of Selin Co was obtained from Hydroweb (https://hydroweb.next.theia-land.fr). GRACE data used in this study can be accessed from https://www2.csr.utexas.edu/grace/RL06_mascons.html.

**Supplement.** The supplement related to this article is available online at XXX.

**Author contributions.** G. Zhang designed the concept of this study. F. Xu drafted the manuscript. All authors reviewed and edited the final manuscript.

**Competing interests.** The authors declare no conflicts of interest relevant to this study.

**Acknowledgements.** We thank Olivier Dewitte for the professional handling of the manuscript and Adam Emmer and an anonymous reviewer for their constructive comments, which improved the manuscript.

**Financial support.** This study was supported by grants from the National Natural Science Foundation of China (Grant No. 42301150), the China Post-Doctoral Program for Innovative Talents (Grant No. BX20230387), the Second Tibetan Plateau Scientific Expedition and Research Program (2019QZKK0201), and the Basic Science Center for Tibetan Plateau Earth System (BSCTPES, NSFC project No. 41988101-03). RIW was supported by the UK Research and Innovation (UKRI) Natural Environment Research Council (NERC): Independent Research Fellowship [grant number NE/T011246/1 and grant reference number NE/X019071/1, "UK EO Climate Information Service".

**Review statement.** This paper was edited by Olivier Dewitte and reviewed by Adam Emmer and an anonymous referee.

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
