# Peer review of "Recent inland large lake outbursts on the Tibetan Plateau: Processes, causes and mechanisms"

_Natural Hazards and Earth System Sciences, 2024_

## Author Comment (AC1)

**#Reviewer 1**

This study reports three examples of lake outbursts on the Tibetan Plateau. Reported outbursts involved huge amounts of water, threatened infrastructure and are worth reporting considering the implications for future outburst hazards in the region. The text is in general well-written and the manuscript is accompanied by rich figures of high quality and information content. However, it is bit confusing for the readers that the authors describe two outbursts (from Lake Zonag in 2011 and from Lake Selin Co in 2023) in some parts of the study, while the study also contains the information about another one from Lake Donggei Cuona in 2024. To make the study clearer, I suggest the authors to introduce separate study area section (including brief description of three GLOF sites described further in the text) and to describe the studied GLOFs in separate sub-sections of the Results. I also recommend to revise discussion section, put reported outbursts in the context of other outburst in the region and present solid implications of this research for understanding outburst occurrence there. Detailed suggestions / comments / questions are provided below:

**Response:** Thank you for your careful review of our manuscript. After a thorough understanding of the reviewer's comments, we have carefully addressed each point raised. Of these, we have added the subsection in "2 Data and methods" section to describe three GLOF sites and have described the examined GLOFs in separate sub-sections of the Results. Meanwhile, we have enhanced the discussion of inland lake outburst with other GLOFs documented in the TP. Below, please find our detailed responses to the original feedback. For clarity, the original reviewers' comments are presented in black, and our responses are shown in blue.

L25: the 2011 GLOF has not accompanied the 2023 peak – please reword

**Response:** Thanks, we have improved this sentence as "_Here, a long-term satellite lake mapping shows that the number and surface area of lakes on the Tibetan Plateau exhibit an increased trend over the past 50 years, peaking in 2023. Two notable outburst events occurred during this period: Zonag Lake (~150 km$^2$ in 2023) on 15 September 2011 and Selin Co (~2,465 km$^2$ in 2023, the largest lake in Tibet) on 21 September 2023._".

L29: Gt is not the unit of volume

**Response:** Thanks, you are right, and Gt is the unit of mass. Therefore, we have improved this sentence as "_The Selin Co outburst resulted in a water mass loss of ~0.3 Gt, the downstream Bange Co experienced a water level rise of ~2.3 m and an area expansion of ~18%._".

L38: vertical motion of what?

**Response:** We mean vertical motion of moisture. We have improved this sentence as "_For Zonag Lake, thermodynamic effects, i.e. changes in the atmospheric moisture, are the most important, while for Selin Co, dynamical effects, i.e. the vertical moisture motion induced by the changes in atmospheric circulation, dominate the precipitation patterns._".

L44: I suggest not to describe plateau lakes as alpine lakes (plateau environment and alpine environment differ in my understanding)

**Response:** We thank the reviewer for this suggestion and have revised "alpine lakes" to "plateau lakes".

Fig. 1: you might mention that 1970 – 2018 data come from Zhang et al., 2019

**Response:** We thank the reviewer for this suggestion and have added the sentence "_Lake data from 1970 to 2018 are from Zhang et al. (2019a)._" in the Fig.1 caption.

L78: please provide a reference

**Response:** We have added a related reference as follow.

Reference: Xu, F., G. Zhang, R. I. Woolway, K. Yang, Y. Wada, J. Wang, and J.-F. Crétaux (2024), Widespread societal and ecological impacts from projected Tibetan Plateau lake expansion, Nature Geoscience, 17(6), 516–523.doi: 10.1038/s41561-024-01446-w.

L80: you actually describe three throughout the text

**Response:** We mainly analyzed Zonag Lake event on 15 September 2011 and Selin Co event on 21 September 2023 because of their widespread impacts, whereas the Donggei Cuona Lake event was only briefly discussed, as a dam at its outlet has subjected the lake to human intervention, and its outburst duration was relatively short. To avoid confuse, we have improved this sentence as "*In this study, we focus on two notable inland lake outburst events: Zonag Lake on 15 September 2011 and Selin Co on 21 September 2023. The Donggei Cuona Lake event on 18 February 2024 is briefly discussed due to the human management of the lake and the relatively short duration of the outburst.*".

L82: field surveying is mentioned here but not described in methods and data section – please provide more details or delete it from here

**Response:** Thanks for your suggestion, we have added the description about field surveying in the section "2.7 Field surveying" as follow:

"*2.7 Field surveying*

*The field surveying was conducted in October 2023 and August 2024 for Selin Co outburst event. During the field surveying, we examined the breach, the flood channel, and damaged roads, and inlet of Bange Co. Uncrewed aerial vehicle (UAV) was used to capture the orthophotos of flood river channel from Selin Co to Bange Co, breach, as well as road and settlements at risk. Then, high accuracy DEM was generated from UAV orthophotos using Agisoft Metashape and Context Capture software to capture the depth and extent of Selin Co flood inundation.*"

L116: this is first time you mention this lake and outburst (see my general comment)

**Response:** Thanks for your suggestion, we have added the description about examined lakes in the section "2.1 Study area" as follow:

"*2.1 Study area*

*Zonag Lake is situated in the north of Hoh Xil National Nature Reserve in Qinghai Province and the Sanjiangyuan National Park, where a large number of Tibetan antelopes are lambing, known as "Tibetan antelope delivery room" (Chen et al., 2021b). On 15 September 2011, Zonag Lake burst, forming a connected hydrological system with Kusai Lake, Hedin Noel Lake and Yanhu Lake. Zonag-Yanhu Lake basin covers a total area of ~8,564 km², with an extensive distribution of permafrost and periglacial landforms. The average elevation is 4600 m a.s.l. Based on nearby Wudaoliang meteorological station data from 1961 to 2019, the annual mean temperature is about -5.1℃ and the annual mean precipitation is 299.8 mm (Chen et al., 2021b). The region is characterized by three vegetation types: alpine meadows, grasslands and deserts (Liu et al., 2019).*

*Selin Co is the largest lake in Tibet and the second largest in the TP, with an area of ~2,465 km² in 2023. On 21 September 2023, an outburst occurred at Selin Co, resulting in flooding downstream into eastern Bange Co. The total area of Selin Co basin is ~44,437 km², covering 369.7 km² of glacier and 13,404 km² of permafrost (Wang et al., 2022b). Selin Co is fed by four major rivers: Zhajiazangbu, Boquzangbu, Alizangbu and Zhagenzangbu, and has several upstream exorheic lakes, including Qiagui Lake, Wuru Lake and Cuoe Lake. The basin is dominated by a cold, semi-arid monsoon climate, with an average temperature of around 0 °C and average annual precipitation of ~350 mm (Tong et al., 2016). Donggei Cuona Lake is located in the western part of TP (98.55° E, 35.28° N), with an area of ~260 km² in 2023. Donggei Cuona Lake is an exorheic lake with a dam at the outlet. On 18 February 2024, a breach occurred in the left sluice chamber of Donggei Cuona Lake, causing water to flow out of the lake."*

L152: please provide more details how did you set this value? Based on field measurements?

**Response:** Thanks. Some of the previous parameters were based on assumptions and now we determined them based on field measurements and remote sensing observations. After updating the parameters, we rerun the model and the simulations were good. We have added more details about parameter as: "*The width and breach formation time of the breach was determined by tracking its progression using high-resolution PlanetScope imagery. Specifically, the breach width was approximately 60 m on 23 September 2024, expanded to ~160 m by 27 September, reached ~180 m by 3 October, and finally developed to ~200 m on 8 October. Based on this progression, we estimated that the breach formation time took 18 days beginning from the initial breach on 21 September and width of breach was 200 m. The depth of the breach was set to 2 m, which is based on field measurements (Lei et al., 2024). The total calculation time was set from 21 September to 13 October 2023. The Manning roughness coefficient was set to 0.04 based on field surveys, which shows that the river channel area is covered by sparse herbaceous vegetation.*".

L158-160: what is the justification of this assumption?

**Response:** Due to the lack of the continuous observations, there is uncertainty in the discharge fitted by the spline method. Therefore, we have removed this continuous spline-fitted discharge and kept only the mean discharge over a period calculated from the PlanetScope observation and hypsometric curve.

L163: I suggest to distinguish outburst drivers (e.g., climate change) and mechanisms (dam breaching)

**Response:** Thank you for your suggestion. While we acknowledge the importance of distinguishing between outburst drivers and mechanisms, the primary focus of our analysis is on the factors influencing lake outbursts, such as long-term precipitation increase and other related environmental drivers. The atmospheric mechanisms we analyzed mainly explain how increased precipitation contributes to lake expansion. Regarding the dam breaching mechanism, we interpret it as a natural result of the lake expanding to the maximum extent that the surrounding terrain can accommodate. Therefore, dam breaching occurs as a consequence of the lake's physical expansion, which is different from dam breaching for glacial lake, emphasizing dam stability or failure processes. Given the focus of our study, we have chosen to emphasize the atmospheric drivers.

L181: please consider moving this section before 2.4

**Response:** Thanks, we have moved this section forward.

L230: representativeness for what?

**Response:** The representativeness means representation of the natural expansion of lakes on the Tibetan

Plateau. We have improved this sentence as follow: *"Here, we focus on the outburst events in Zonag Lake and Selin Co due to their wide-ranging impacts (i.e., damage to infrastructure, ecosystems, and biodiversity) and representativeness of the natural expansion of lakes on the TP."*.

Fig. 3: since the study highlights the impact of precipitation increase on lake evolution, I suggest to plot lake areas shown in parts a and c against precipitation data from Fig. 7. For part b, please consider adding arrows and description directly in photos, so it is easier to understand what they illustrate

**Response:** We thank the reviewer for this suggestion. We have displayed the lake area in Fig.3a and 3c with precipitation change in Fig.7 in one subplot to facilitate comparison (Below). Also, for Fig.3b, the arrows and descriptions have been added to facilitate understanding.

[Figure]

**Figure 3.** The changes and impacts of Zonag Lake and Selin Co. (a) The area change of Zonag Lake and downstream Kusai Lake, Haidingnuoer and Yanhu Lake, as well as the annual precipitation change from Wudaoliang weather stations (Blue). (b) The impact of Zonag Lake outburst, including the formation of new channel, soil erosion and the disruption of Tibetan antelope migration routes. (c) The area changes of Selin Co, as well as the annual precipitation change from Shenzha weather stations (Blue). The inset shows the rapid expansion of Selin Co and Bange Co in 2022-2024. (d) The impact of Selin Co outburst, including formed

breach, damaged road, new channel between Selin Co and adjacent Bange Co. Selin Co has also approached near the road on the south-west side (lower right panel).

L258: "water had reached he road" rather than "had accumulated near"?
**Response:** Thanks. We mean an accumulation of precipitation, not the outburst flood of Selin Co, which has not yet occurred. To avoid the confuse,we have improved this sentence as followed "*Selin Co was still in normal condition on 16 September, whereas by 20 September there was already a substantial amount of water near the S208 road indicating signs of heavy precipitation.*".

L259: what signs of heavy precipitation?
**Response:** Based on high-resolution Planet remote sensing imagery, it was found that Selin Co had not yet breached on 20 September 2024, but the road was heavily waterlogged, which implied that heavy precipitation was occurring.

L265; the precision of discharge to two decimal places is inappropriate considering uncertainties and assumptions of model inputs (including input hydrograph)
**Response:** Thanks, we have revised two decimal places to four decimal places for small discharge of Selin Co for this sentence.

Fig. 4: "The outburst" instead of "outbursting process"?; i and j – km instead of m on x axis?
**Response:** Corrected.

L286: how reliable is this number considering DEM resolution, accuracy and uncertainties of modelling?
**Response:** We thank the reviewer for this suggestion. To validate the reliability of the modelling, we conducted three verification actions. First, since all the floodwaters flowed into Bange Co, we compared the simulated water level change in Bange Co with satellite-based measurements. The model simulated a water level rise of 2.23 m, which matches well the observed increase of 2.30 m. Second, the simulated flood path and extent showed good spatial consistency with PlanetScope imagery. The extent of Bange Co observed on 13 October 2024 from PlanetScope imagery also aligned well with the simulated extent change. Third, we compared the simulated flood depth with UAV-derived DEM measurements of the post-breach flood channel. The model simulated flood depth at the right side the damaged road was 1.63 m, while UAV measurements indicated a depth of 1.96 m (the elevation difference between the riverbank and riverbed), showing good agreement. Therefore, the simulated flood process has good reliability. We have added more details as: "*To validate the reliability of the model, we implemented validation in three aspects. First, we compared the simulated water level changes in Bange Co with satellite-based measurements, as all floodwaters flowed into the lake. Second, the simulated flood path and extent were verified by comparing them with PlanetScope imagery to assess spatial consistency. Third, the simulated flood depth was validated with UAV-derived DEM measurements (the elevation differences between the riverbank and the riverbed can reflect the inundation depth of flood) of the post-breach flood channel.*" in method section, and "*A water level rise of 2.23 m was identified from the hydrodynamic simulation, which matches well the observed increase of 2.30 m. Additionally, the simulated flood path and extent as well as the boundary of Bange Co showed good spatial consistency with PlanetScope imagery. We also compared the simulated flood depth with UAV-derived DEM measurements of the post-breach flood channel. The model simulated flood depth at the right side the damaged road was 1.63 m, while UAV measurements indicated a depth of 1.96 m, showing overall good agreement (Figure 7).*" in "3.2.2 Selin Co outburst event"

subsection.

[Figure]

**Figure 7**. The validation of the Selin Co outburst flood simulation. (a) Comparison of the simulated flood path with PlanetScope observations of Selin Co on 13 October 2024. (b) Comparison of Bange Co boundary with PlanetScope observations of Selin Co on 13 October 2024. (c) The UAV-derived DEM of the Selin Co flood path. (d) The UAV-DEM near the damaged road, and the inset shows the elevation profile of the right side of the damaged road to indicate the flood inundation depth.

L288-293: the distance between the two lakes (according to Fig. 4) is about 12 km. This gives average flood velocity < 1 km per hour. This is very slow, definitely not rapid as described here.
**Response:** We thank the reviewer for this suggestion. We have improved this sentence as follow: "*This suggests that even in flat areas, the impact of flooding can be significant in a short period of time and requires the establishment of an early warning system*".

L292: agree about what?
**Response:** We mean "agree with the flood inundation extent from remote sensing observations (Fig.5 and Fig. 7).

L299: is this the case in the TP?
**Response:** For endorheic lakes on the Tibetan Plateau, this is the case. The two recorded outburst events from endorheic lakes in the region (Zonag Lake and Selin Co), as mentioned in this study, both involved lakes that expanded until they overflowed, leading to erosion at the breach site. Of these, it can be noted from the Selin Co event that the width of the breach widened from about 60 m on 23 September 2024 to 200 m as a result of erosion. However, this mechanism does not apply to glacial lakes. Glacial lake outburst floods (GLOFs) are typically triggered by external forces such as ice avalanches or landslides, which cause rapid destabilization. In contrast, endorheic lake outbursts are more internally driven, as the lakes expand naturally due to increased

water input, eventually exceeding the terrain's capacity to contain them. When overflow occurs in endorheic lakes, it often initiates erosion of the dam, potentially turning overflow into a full breach. Therefore, while the risk of overflow and erosion is pertinent to endorheic lakes, it is less relevant for glacial lakes, where external triggers are the primary drivers of outburst events.

L300: how did you get this discharge? It is written in the >Methods that only the outburst of Selin Co was modelled. The 2,191 m3/s average discharge over 28 days gives outburst volume exceeding 5 km^3 (!!). Íf so, this is among the largest (maybe even the largest) outburst floods in the Anthropocene and it should be highlighted

**Response:** Thank you for your comment. Upon reviewing the calculation, we have identified a minor discrepancy in the originally reported discharge. The discharge was calculated by dividing the total outburst water volume by duration. Based on changes in lake area (a decrease of 107.5 $km^2$) and water level (a drop of approximately 25.59 m over 28 days), the total water volume loss was re-estimated using volume estimation formula (Eq 1), resulting in a total water loss of 5.42 Gt ($km^3$). This corresponds to an average discharge of 2,238 $m^3$/s over the 28-day period, slightly higher than the previously reported value of 2,191 $m^3$/s. We will revise the manuscript to reflect the corrected value.

$$\Delta V = \frac{1}{3} \Delta h \times (A_1 + A_2 + \sqrt{A_1 \times A_2})$$
(1)

where $\Delta V$ is the lake volume change ($km^3$), which is usually converted to water storage change in Gt (gigaton) to compare water mass gain or loss; A1 and A2 are lake areas ($km^2$) at stage 1 and stage 2, respectively; $\Delta h$ is the water levels (m) at the corresponding stages.

Fig. 7: please add a trendline for part a and consider plotting this against lake area (see my suggestion above)
**Response:** We thank the reviewer for this suggestion. We have added a trendline for original Fig.7a (now Fig.3)

[Figure]

**Figure 3.** The changes and impacts of Zonag Lake and Selin Co. (a) The area change of Zonag Lake and downstream Kusai Lake, Haidingnuoer and Yanhu Lake, as well as the annual precipitation change from Wudaoliang weather stations (Blue). (b) The impact of Zonag Lake outburst, including the formation of new channel, soil erosion and the disruption of Tibetan antelope migration routes. (c) The area changes of Selin Co, as well as the annual precipitation change from Shenzha weather stations (Blue). The inset shows the rapid expansion of Selin Co and Bange Co in 2022-2024. The inset shows the rapid expansion of Selin Co and Bange Co in 2022-2024. (d) The impact of Selin Co outburst, including formed breach, damaged road, new channel between Selin Co and adjacent Bange Co. Selin Co has also approached near the road on the south-west side (lower right panel).

L312: I was also wondering whether is there possibly any substantial contribution from melting glaciers? Please discuss this

**Response:** The contribution of glacier melt to lake expansion is relatively small, with precipitation being the dominant factor. For Selin Co, DEMs derived from KH-9 and CoSSC-TanDEM-X data estimate glacier contributions to lake expansion at approximately 3.5% to 16.3% (Chen et al., 2021). Similarly, degree-day model simulations from 1976 to 2013 suggest a contribution of about 10% (Tong et al., 2016). For Zonag Lake,

model simulations indicate that glaciers and snow contribute around 21% to lake expansion (Wang et al., 2022). Therefore, glacier melt plays a relatively minor role in the overall lake expansion process.

**References**:

Chen, W., Yao, T., Zhang, G., Li, S., and Zheng, G.: Accelerated glacier mass loss in the largest river and lake source regions of the Tibetan Plateau and its links with local water balance over 1976–2017, Journal of Glaciology, 67, 577-591, 10.1017/jog.2021.9, 2021.

Tong, K., Su, F., and Xu, B.: Quantifying the contribution of glacier meltwater in the expansion of the largest lake in Tibet, Journal of Geophysical Research: Atmospheres, 121, 10.1002/2016jd025424, 2016.

Wang, L., Liu, H., Zhong, X., Zhou, J., Zhu, L., Yao, T., Xie, C., Ju, J., Chen, D., and Yang, K.: Domino effect of a natural cascade alpine lake system on the Third Pole, PNAS nexus, 1, pgac053, 2022.

L330: this extreme precipitation prior the outbursts is not shown

**Response:** Thank you for this comment. We have added a subplot in original Figure 7 (now Figure 4) to show the daily precipitation changes prior to the outburst based on data from a nearby weather station, which indicate the continuous heavy precipitation prior to the outburst (Figure 4).

[Figure]

**Figure 4 (original Figure 7).** The change in extreme precipitation derived from Shenzha (near Selin Co) and Wudaoliang (near Zonag lake) weather stations. (a) Extreme precipitation (total precipitation that exceeds a 95th and 99th percentile during the historical period from 1981 to 2010) change from 1980 to 2023. (b) The daily change of precipitation prior to the outburst. (c) Monthly extreme precipitation change based on the 95th percentile from 1980 to 2023. (d) Monthly extreme precipitation change based on the 99th percentile from 1980 to 2023. The location of the weather station was shown in Figure 1.

Figs. 8-11: these figures are difficult link to outbursts of the two lakes. Maybe one synthesizing figure can be presented in the main manuscript and the rest goes to the supplement?

**Response:** We appreciate the reviewer's suggestion. We would like to retain original Fig. 8 in the main manuscript as it serves as a summary figure. Original Figs. 9 and 10, which provide detailed analysis of different contributions, have been moved to the supplementary. However, due to the important role of atmospheric circulation on water vapor increase, we believe original Fig. 11 is important for understanding the overall process and would like to keep it in the main manuscript.

L411: how does it accelerated permafrost degradation?

**Response:** Using SBAS-InSAR analysis of Envisat and Sentinel-1 datasets, accelerated permafrost degradation around Yanhu Lake since 2014 was observed. This degradation is likely linked to alteration in the thawing-freezing cycles and melting of ground ice, which may have impacted hydrological connectivity and soil permeability (Lu et al., 2020). The changes in surface thermal conditions following the lake outburst may have further contributed to these processes.

**Reference**:

Lu, P., Han, J., Li, Z., Xu, R., Li, R., Hao, T., and Qiao, G.: Lake outburst accelerated permafrost degradation on Qinghai-Tibet Plateau, Remote Sensing of Environment, 249, 112011, 10.1016/j.rse.2020.112011, 2020.

L429: the recommendations mentioned in this section are general; to make stronger point, reported events should be put in the context of other GLOFs documented from the TP (in terms of outburst timing, drivers, etc.)

**Response:** We thank the reviewer for this suggestion. We have added the description about the discussion with other recorded GLOFs in the TP as follow:

"*Lake outburst events on the TP can be divided into two categories based on their driving mechanisms: endorheic lake outbursts and GLOFs. The two documented endorheic lake outburst events in this region, Zonag Lake and Selin Co, both involved lakes expanding due to increased water input until they overflowed. This overflow led to erosion at the breach site, eventually resulting in full outbursts. For instance, in the Selin Co event, the breach widened from ~ 60 m on 23 September 2024 to ~200 m on 8 October 2024 due to ongoing erosion, which illustrates how overflow can rapidly escalate into a breach. In contrast, GLOFs are typically triggered by external factors such as ice avalanches and landslides. These events cause rapid destabilization of glacial lakes, leading to sudden, high-discharge floods that peak within hours or cause the end of flood releases. GLOFs tend to be more frequent due to the unstable nature of glaciers and other factors. For example, the Gongbatongsha Co outburst events in 2016 in China demonstrate the devastating potential of these sudden releases (Wang et al., 2024), which are often difficult to predict and have higher peak discharge compared to endorheic lake outbursts.*

*While both types of events pose significant risks, endorheic lake outbursts are more gradual in nature, driven by the internal dynamics of lake expansion. In contrast, GLOFs are driven by external forces, often resulting in more abrupt and catastrophic flooding. Understanding these differences is critical for risk assessment and management on the TP, as the two types of lakes require different monitoring and mitigation approaches. For instance, monitoring glacier dynamics and potential ice avalanches is crucial for GLOFs, whereas tracking lake water levels and identifying potential overflow points is key for managing endorheic lake outbursts.* "

**Reference**:

Wang, X., Zhang, G., Veh, G., Sattar, A., Wang, W., Allen, S. K., Bolch, T., Peng, M., and Xu, F.: Reconstructing glacial lake outburst floods in the Poiqu River basin, central Himalaya, Geomorphology, 449, 10.1016/j.geomorph.2024.109063, 2024.

To sum up, I'm convinced this study fits well in the journal and would be of interest for the readers. I recommend revisions of the structure, introduction of a separate study area section and revision of the discussion section (moderate to major revisions).

**Response:** We sincerely thank the reviewer for the careful and constructive comments. We agree that these

suggestions will enhance the overall quality of our manuscript. These changes have been made to ensure the manuscript is clearer and more accessible to the readers.

---

## Author Comment (AC2)

**General comments:**

The manuscript presents interesting case studies that contribute to the understanding of outburst processes and introduces valuable data that could enhance knowledge of natural hazards. However, the depth of the presentation and interpretation is somewhat lacking. Conclusions are often drawn too quickly, with the authors frequently suggesting the need for further research without fully engaging with the existing data. This limits the overall impact of the study.

The explanation of methods and the presentation of results are imprecise. Some data and results, particularly in Section 3.3, are introduced abruptly, and the connection to broader findings is not always clear. Moreover, the discussion is superficial, leaving the reader with unanswered questions about the mechanisms being studied.

A significant issue lies with the figures, which, while containing essential information and most of the results presented, are often difficult to interpret. The explanations provided are too brief. Clearer, more detailed descriptions and better-structured visual aids are needed to improve the flow of information and help readers understand the material presented. Additionally, numerous textual errors and a lack of precision in technical details reduce the manuscript's clarity and understanding, weakening its overall impact.

**Response:** We sincerely thank the reviewer for the constructive comments, which has greatly contributed to the improvement of our manuscript. In response to the comments, we have carefully revised the manuscript to strengthen the depth of our analysis. To address concerns about drawing quick conclusions, we included additional evidence and detailed reasoning to support our conclusions.

To improve clarity in the methods and result sections, we have added more detailed explanations of the methodologies used, ensuring a clearer understanding of our approach. Additionally, in Section 3.3, we have revised the title to "3.3 Climate mechanisms of recent lake outbursts" and strengthened the presentation of climate mechanisms to make clear the purpose and necessity of examining the atmospheric mechanisms of precipitation. This change improves the relevance to earlier sections of the manuscript. The discussion has been expanded to include comparisons between the outburst events discussed in our study and other type events on the Tibetan Plateau. This comparative analysis highlights the distinct causes and responses of different outburst events, enriching the overall context.

Furthermore, we have made substantial revisions to the figures, improving their clarity and interpretability. These changes include reorganizing certain figures, adding clearer labels, and expanding the figure captions to provide more comprehensive explanations. Additionally, we have added new figures to better support the findings and enhance the manuscript's overall readability. We have carefully read the text to eliminate any technical errors and ensure greater precision throughout the manuscript. We appreciate the reviewer's valuable suggestions, which have been helpful in refining and improving our work. Below, please find our detailed responses to the original feedback. For clarity, the original reviewers' comments are presented in black, and our responses are shown in blue.

**Specific comments**

L46-48: Why are alpine lakes considered sentinels of climate change?

**Response:** Alpine lakes, located in climate-sensitive regions (such as the Tibetan Plateau), are directly influenced by climate factors such as glacier/snow melt, precipitation, and evaporation, making their response to climate change rapid and significant. Additionally, the relatively closed ecosystems of alpine lakes make it easier to observe the impacts of climate change on lakes. Furthermore, as long-term monitoring targets, alpine lakes provide direct evidence of climate change through recorded changes in sensitive indicators such as lake

area, water level, ice cover duration, and surface water temperature. Therefore, they are regarded as sentinels of climate change. Some related papers below also support this.

Zhang, G., and S. Duan (2021), Lakes as sentinels of climate change on the Tibetan Plateau, All Earth, 33(1), 161-165, doi: 10.1080/27669645.2021.2015870

Adrian, R., C. M. O'Reilly, H. Zagarese, S. B. Baines, D. O. Hessen, W. Keller, D. M. Livingstone, R. Sommaruga, D. Straile, and E. Van Donk (2009), Lakes as sentinels of climate change, *Limnol Oceanogr*, *54*(6), 2283–2297, doi: 10.4319/lo.2009.54.6_part_2.2283

L107-L116: It is not very clear how the PlanetScope imagery was used. Was it employed as orthophoto for visual interpretation and lake delineation, or was it to compute NDWI using the NIR and Green bands?

**Response:** In our study, the PlanetScope imagery was used to track the outburst process of Selin Co by visual interpretation, and to extract the area of Selin Co during 2022-2023 based on the NDWI index. To be clearer, we have improved the sentence as "*High-resolution PlanetScope satellite images with 3-m pixel and a 1-day cycle, were used to track the lake outburst process of Selin Co by visual interpretation, and to map the area change of Selin Co during 2022-2023 based on the NDWI index.*".

L133: "The area-water level relationship was used to obtain water levels from the 1970s to 2023." It's not clear what relationship the authors are referring to here.

**Response:** The area-water level relationship means the significant linear relationship between lake area and water level. Based on this relationship, we can estimate the area change by inputting water level, or water level change by inputting area. For example, LELVEL=0.015*AREA+4512.749 is the relationship between water level and area in Selin Co. We have improved this sentence as: "*The area-water level linear relationship was used to obtain water levels from the 1970s to 2023 by inputting lake area (Zhang et al., 2021)*".

L118-134: If I understood correctly, they calculated water levels using Sentinel-3 for the period ranging from 2016-2024 and using CryoSat-2 for 2010-2020? Did the authors intend to have this overlap? Or did I miss something? This section (2.2) could be slightly rearranged to improve readability.

**Response:** Thank you for your careful comment. CryoSat-2 data was used in previous study but not this. We apologize for the confuse in the text, and we have removed the mention of CryoSat-2 to clarify this.

L148: What is the resolution of NASADEM? Is it relevant to use that data source to define the depth of the breach for modeling?

**Response:** The spatial resolution of NASADEM is 30 m, but it is not relevant to determining the breach depth. The previous breach depth was estimated based on the assumption of a gentle slope at the breach site. However, the current breach depth is based on field measurements from Lei et al. (2024).

**Reference**:

Lei, Y., J. Zhou, T. Yao, B. W. Bird, Y. Yu, S. Wang, K. Yang, Y. Zhang, J. Zhai, and Y. Dai (2024), Overflow of Siling Co on the central Tibetan Plateau and its environmental impacts, Sci Bull. doi: 10.1016/j.scib.2024.07.035.

L152-153: How did the authors estimate the depth of the breach? It's only written "due to the flat topography," but I'm wondering how they concluded it was 0.8m?

**Response:** The previous breach depth was estimated based on the assumption of a gentle slope at the breach

site. However, the current breach depth (2 m) in the revised manuscript is based on field measurements from Lei et al. (2024). After all the parameters were updated, we rerun the model and have improved this sentence as: "_The depth of the breach was set to 2 m, which is based on field measurements (Lei et al., 2024)_".

**Reference**:
Lei, Y. et al., 2024. Overflow of Siling Co on the central Tibetan Plateau and its environmental impacts. Sci Bull. doi:10.1016/j.scib.2024.07.035.

L153: How did they determine the breach formation time? Were there any eyewitness accounts?

**Response:** We thank the reviewer for this question. The breach formation time was initially based on assumptions but is now accurately determined using high-resolution PlanetScope imagery in the revised manuscript. Specifically, the breach width was approximately 60 m on 23 September 2024, expanded to ~160 m by 27 September, reached ~180 m by 3 October, and finally developed to ~200 m on 8 October. Based on this progression, we determined that the breach formation time took 18 days, beginning from the initial breach on 21 September. After all the parameters were updated, we rerun the model. We have improved this sentence as: "_The width and breach formation time of the breach was determined by tracking its progression using high-resolution PlanetScope imagery. Specifically, the breach width was approximately 60 m on 23 September 2024, expanded to ~160 m by 27 September, reached ~180 m by 3 October, and finally developed to ~200 m on 8 October. Based on this progression, we estimated that the breach formation time took 18 days beginning from the initial breach on 21 September and width of breach was 200 m_".

L154: Based on which parameters did the authors set the Manning roughness to 0.04?

**Response:** The Manning roughness coefficient was set to 0.04 based on field surveys, which showed that the river channel area is covered by sparse herbaceous vegetation. We have improved this sentence as: "_The Manning roughness coefficient was set to 0.04 based on field surveys, which showed that the river channel area is covered by sparse herbaceous vegetation._".

L184: What do the authors mean by "long- and short-changes"?

**Response:** The "long-changes" refer to precipitation changes over several decades, while "short-changes" indicate precipitation variations occurring shortly before the lake outburst event. We have clarified these terms in the revised manuscript as "_The daily in situ precipitation records from the China Meteorological Administration (https://www.cma.gov.cn) were used to the analyze long changes overs several decades and precipitation variations occurring shortly before the lake outburst event_".

L233: Is there any measurement of Zonag Lake's area between August 22, 2011, and September 15, 2011?

**Response:** Yes, the area of Zonag Lake in August 22, 2011 and September 15, 2011 is from the Chinese HJ-A/B satellites satellite observation (Liu et al., 2016). These satellite measurements provide accurate data for assessing the lake's changes during this period. We have improved the sentence as "_The lake area of Zonag Lake reduced by ~107.52 km2 observed by Chinese HJ-A/B satellites satellite within 28 days (~3.84 km2/day) (Liu et al., 2016)_".

**Reference**:
Liu, B., Y. e. Du, L. Li, Q. Feng, H. Xie, T. Liang, F. Hou, and J. Ren (2016), Outburst Flooding of the Moraine-Dammed Zhuonai Lake on Tibetan Plateau: Causes and Impacts, IEEE Geoscience and Remote Sensing

Letters, 13(4), 570-574.doi: 10.1109/lgrs.2016.2525778.

Figures 3i and 3j: What are they used for? What do they illustrate? I'm not sure how they contribute to the interpretation.

**Response:** Figures 3i and 3j illustrate the impact of the outburst event. To improve clarity, we have added the description and arrows to the panels to better facilitate understanding. The figure was referenced in the subsection "4.2 Consequences of Zonag Lake and Selin Co outburst" to highlight the specific effects of the outburst.

In Figure 4, I'm not sure how the profile P1-P2 is relevant.

**Response:** The profile P1-P2 illustrates the elevation change along the river channel between Selin Co and Bange Co before the Selin Co outburst. This profile is relevant as it shows that the water level in Selin Co had risen to the maximum level that the basin could contain, providing important context for the outburst event.

Figures 6g and 6h: It might be interesting to add the location of the built dam. I think I misunderstood something because, regarding Figures 6e and 6g, it seems that the water depth increased between September 28 and October 13, but the inset shows no variation.

**Response:** Thank you for the suggestion. We have added the location of the built dam in original Figures 6g and 6h (now Figures 8g and 8h). The water depth of Bange Co increased between September 28 and October 13 due to the inflow from the Selin Co outburst flood. However, the inset shows the water depth at the damaged road, not in Bange Co.

Figures 7a and 7b: What about the peaks in 2008 and 2002, respectively, for Selin Co and Zonag Lake? Did they induce any hazards in the region?

**Response:** Original Figures 7a and 7b (now merge them into Figure 3a, b) show the interannual variation in precipitation, with peaks in 2008 for Selin Co and 2002 for Zonag Lake representing sudden increases in annual precipitation observed at nearby stations. These peaks contributed to the expansion of the lakes but did not induce any hazards, as the lakes had not yet reached the maximum extent to trigger an outburst during this period.

L316-317: The authors stipulate, "In addition, seismic events prior to the breach may have weakened the geological stability of the dam (Liu et al., 2016)." I found this statement a bit vague. I would prefer some specifics about these events (location, intensity, impact).

**Response:** Thank you for the suggestion. We have added specific descriptions for the seismic events as: "*Additionally, two seismic events occurred prior to the breach: one on 27 July 2011 (62 km from Zonag Lake, magnitude 4.0), and another on 22 August 2011 (57 km from Zonag Lake, magnitude 3.1). These events may have weakened the geological stability of the lake dam (Liu et al., 2016)*".

Figures 8e and 8f need to be described in more detail for non-expert readers (the explanation of the x-axis comes in Figure 9, but it should be detailed earlier).

**Response:** Thank you for your suggestion. We have added more detail in caption as "*(e, f) The contribution of each term (precipitation ($P'$), Evaporation ($E'$), the change in vertical ($-\langle\omega\partial_p q\rangle'$) and horizontal moisture advection ($-\langle V_h \cdot \nabla_h q\rangle'$), residual ter effects m ($\delta$), thermodynamic effects $-\langle\bar{\omega}\partial_p q'\rangle$, dynamic ($-\langle\omega'\partial_p\bar{q}\rangle$), the nonlinear effects ($-\langle\omega'\partial_p q'\rangle$)) of moisture budget components (mm/day) to precipitation changes averaged over*

*the Zonag Lake and Selin Co basins.*"

L330: I don't clearly see the "continuous heavy or extreme precipitation prior to the outburst" mentioned by the authors.

**Response:** Thank you for the question. We have added a subplot in original Figure 7 (now Figure 4) to show the daily precipitation changes prior to the outburst based on data from a nearby weather station, which indicate the continuous heavy precipitation prior to the outburst (Figure 4).

[Figure]

**Figure 4 (original Figure 7).** The change in extreme precipitation derived from Shenzha (near Selin Co) and Wudaoliang (near Zonag lake) weather stations. (a) Extreme precipitation (total precipitation that exceeds a 95th and 99th percentile during the historical period from 1981 to 2010) change from 1980 to 2023. (b) The daily change of precipitation prior to the outburst. (c) Monthly extreme precipitation change based on the 95th percentile from 1980 to 2023. (d) Monthly extreme precipitation change based on the 99th percentile from 1980 to 2023. The location of the weather station was shown in Figure 1.

L375: "...with the wave train in 2011 being relatively flat and the wave train in 2023 being curved." What does

this observation imply?

**Response:** Thanks. The propagation of wave activity fluxes can induce changes in downstream circulation through the process of energy dispersion. Hence, the propagation path of wave activity flux plays a crucial role in the evolution of downstream circulation patterns. As shown in Figure 10, the propagation path of wave activity flux in 2011 is relatively flat, whereas in 2023, it follows a more curved trajectory. The discrepancies in propagation path of wave activity flux between 2011 and 2023 led to variations in regional atmospheric circulation around the TP.

L395: "Pacific Decadal Oscillation (PDO), and Atlantic Multidecadal Oscillation (AMO)." A few words about this would be appropriate. Also, the authors cite "2023," but is that correct? What is the year 2023 referencing?

**Response:** Thanks. We have added relevant content. "*The decadal increase in precipitation on the TP can be attributed to the external forcing, Pacific Decadal Oscillation (PDO), and Atlantic Multidecadal Oscillation (AMO) (Liu et al. 2021, 2023). For example, the combined influence of external forcing and the PDO could result in an anomalous cyclone over the Inner-TP and a weakened East Asian westerly jet, subsequently contributing to the decadal increase in precipitation in the Inner-TP (Liu et al. 2023). A positive phase of the AMO may lead to a northward shift and weakening of the subtropical westerly jet stream, which in turn affects moisture transport and results in changes in precipitation patterns (Liu et al., 2021; Sun et al., 2020)*".

**References**:

Liu, Y., H. Chen, H. Li, G. Zhang, and H. Wang (2021), What induces the interdecadal shift of the dipole patterns of summer precipitation trends over Tibetan Plateau. International Journal of Climatology. 41(11): 5159-5177.

Liu, Y., H. Wang, H. Chen, Z. Zhang, H. Li, and B. Liu (2023), Anthropogenic forcing and Pacific internal variability-determined decadal increase in summer precipitation over the Asian water tower. npj Climate and Atmospheric Science. 6: 38.

Sun et al., (2020), Why has the Inner Tibetan Plateau become wetter since the mid-1990s? Journal of Climate. 33(19): 8507-8522.

L437-442: "This outburst resulted in a significant increase in discharge pressure in the downstream river channels, which had substantial impacts on human living environments, affecting five townships and resulting in the death of approximately 195 livestock. Additionally, about 24.95 km of roads and pastoral paths and some water management facilities were destroyed by flooding, and some pastures were also inundated. Emergency repairs were promptly undertaken, and the damaged gate was sealed on February 21." Any citation for this information?

**Response:** This information is from CCTV NEWS. We have added the citation in revised manuscript as: "*Wang, H. Flooding in Qinghai Province? Official response: not consistent with the facts did not cause casualties. https://news.cctv.com/2024/03/02/ARTIwBthTZFUklWgPzCIQmOu240302.shtml, CCTV NEWS, last access: 15 October 2024.*"

**Technical corrections**

L52: "The significant expansion of these lakes could potentially threaten the fragile …"

**Response:** Thanks. We have improved this sentence as: "The significant expansion of these lakes could threaten the fragile ecological environment of the TP, particularly by inundating grasslands, altering water resources, disrupting habitats, and impacting biodiversity (Xu et al., 2024a)."

Figure 1: a. The areas corresponding to Zonag Lake basin and Selin Co basin could be colored in a darker grey because they are not very visible as they are. Perhaps the authors could use the same blue as in Fig. 1a-1 and 1a-2. Additionally, orange is used in Fig. 1a-1 to show the diversion engineering, but it shows a newly formed channel in Fig. 1a-2. The authors could consider using another color. "Lake" is missing in the legend of Fig. 1a-1, and I believe it should be "Newly formed channel" and not "formd" in the legend of Fig. 1a-2.

**Response:** Thank you for your helpful suggestions. We have improved by the following: the Zonag Lake and Selin Co basin areas are now colored in the same blue as in Fig.1a-1 and 1a-2 for better visibility. Additionally, the color of the newly formed channel in Fig. 1a-2 has been changed from orange to yellow to avoid confusion with other elements. The missing "Lake" in the legend of Fig.1a-1 has been added, and the term "formd" in Fig.1a-2 legend has been corrected to "formed".

Figure 1a-1 refers to Salt Lake, but it is not mentioned in the text. What about Yanhu Lake mentioned on line 65?

**Response:** The term "Salt Lake" has been corrected to "Yanhu Lake" for consistency with the text.

Figure 1a-2: Correct "formd" to "formed" in the legend.

**Response:** Corrected.

L97-99: The authors could quantify what they meant by "low cloud cover" and specify which months?

**Response:** Thank you for your valuable comment. From the MODIS-observed cloud-cover cycle, it can be found that cloud cover is significantly higher in summer (>50%) compared to autumn and winter. Additionally, lake area extraction in spring and winter are often affected by lake ice cover and snowfall, resulting in inaccurate lake boundaries. For this reason, we prioritize autumn, particularly October, when cloud cover is lower, and the lake area is more stable, reaching its annual maximum. We have added more detail as: "*Based on the MODIS-observed cloud-cover cycle, it can be found that cloud cover is significantly higher in summer (>50%) compared to autumn and winter (Zhang et al., 2017a). Additionally, lake outline extraction in spring and winter are often hampered by frozen ice cover and snow, resulting in inaccurate lake boundaries. Therefore, October images are prioritized due to the low cloud cover and relative stability of the lake area (annual maximum).*"

L100: Use "Green" instead of "GREEN."

**Response:** Corrected.

L107: Since the authors specify the spatial resolution of PlanetScope satellite images, they might also add the spatial resolutions of Sentinel-2 and Landsat-8, which were also used.

**Response:** Thank you for the suggestion. We have added the spatial resolutions of Sentinel-2 and Landsat-8 to the revised manuscript.

The authors could add references regarding the technical specifications of the satellites, as an example: "Since June 2016, over 430+ Doves and SuperDoves sensors from the PlanetScope mission have been launched into 475-525 km sun-synchronous orbits, circling the Earth every 90 minutes." [L108-110], or "Sentinel-3A and Sentinel-3B were launched in February 2016 and April 2018, respectively, both equipped with a dual-frequency (Ku and C-band) Synthetic Aperture Radar Altimeter operating in open-loop mode with a cycle period of 27

days, providing high-quality observations of lake water levels." [L120-123].

**Response:** Thank you for the suggestion. References to the technical specifications of PlanetScope, Sentinel-2, Landsat-8, and Sentinel-3 have been added to enhance the clarity of the methodology.

**References**:

Mullen, A. L., et al. (2023), Using High-Resolution Satellite Imagery and Deep Learning to Track Dynamic Seasonality in Small Water Bodies, Geophysical Research Letters, 50(7). doi: 1029/2022gl102327, 2023. (**for PlaneScope**).

Xu, F., G. Zhang, S. Yi, and W. Chen (2021), Seasonal trends and cycles of lake-level variations over the Tibetan Plateau using multi-sensor altimetry data, Journal of Hydrology, 604, 127251.doi: 10.1016/j.jhydrol.2021.127251. (**for Sentinel-3**).

Yang, X., Q. Qin, H. Yésou, T. Ledauphin, M. Koehl, P. Grussenmeyer, and Z. Zhu (2020), Monthly estimation of the surface water extent in France at a 10-m resolution using Sentinel-2 data, Remote Sensing of Environment, 244. doi: 10.1016/j.rse.2020.111803. (**for Sentinel-2**).

Zhang, G., et al. (2019), Regional differences of lake evolution across China during 1960s–2015 and its natural and anthropogenic causes, Remote Sensing of Environment, 221, 386-404. doi: 10.1016/j.rse.2018.11.038. (**for Landsat**).

Equation 1: What does "Halt" stand for?

**Response:** Thank you for pointing this out. $H_{alt}$ represents the satellite altitude. We have clarified this in the revised manuscript by extending the sentence: "$H_{alt}$ *is the satellite altitude*".

L187: Be consistent in writing dates.

**Response:** We have rechecked the writing dates throughout the text and unified the date format.

L239: Be consistent with the names of the lakes (i.e., Hedin Noel Lake differs in Figure 3a).

**Response:** Corrected.

Figure 3: Due to the size of the second graph focusing on the years 2022 to 2024, Figure 3c appears unclear. Perhaps the readability of this graph could be improved.

**Response:** Thank you for your comment. In Figure 3, we firstly aim to show the long-term variation of Selin Co area from 1975 to 2023, as the lake expansion serves as the foundation for the breach. Additionally, we included an inset focusing on the 2022 to 2024 period to highlight recent changes in Selin Co area. The inset was designed to emphasize short-term variations without losing sight of the broader long-term trend. To further emphasize the changes between 2022 and 2024, we have enlarged the size of the illustration to improve readability.

Figures 4, 5d, and 6: The authors should add a scale and a North arrow.

**Response:** Added.

Typographical Error: "Flood conditions on September 27, 2023" (also correct "September 28" on the left side of the figure).

**Response:** Thanks. We have improved this sentence to "*Flood simulation on 28 September 2023*".

Figure 8a-d: Please add the locations of Zonag Lake and Selin Co.

**Response:** Thanks. We have added a yellow box in Figure 8a-d to represent the locations of Zonag Lake and Selin Co (new Figure 9).

[Figure]

**Figure 9 (original Figure 8)**. The atmospheric mechanism of precipitation-induced events in Zonag Lake and Selin Co. (**a, b, c, d**) Composite maps of anomalous vertically integrated moisture flux based on ERA5 data (WVF; integrated from surface to 100 hPa and from 600 to 100 hPa; vector; kgm$^{-1}$s$^{-1}$) in Zonag Lake basin during 2011 and Selin Co basin during 2023. The shading indicates precipitation anomalies. The black vectors indicate WVF exceeds the reference value. The reference climate state was selected as the average from 1981 to 2010. The yellow boxes represent the location of the Zonag Lake and Selin Co, respectively. (**e, f**) The contribution of each term (precipitation ($P'$), Evaporation ($E'$), the change in vertical ($-\langle\omega\partial_p q\rangle'$) and horizontal moisture advections ($-\langle V_h \cdot \nabla_h q\rangle'$), residual term ($\delta$), thermodynamic $-\langle\bar{\omega}\,\partial_p q'\rangle$, dynamic ($-\langle\omega'\,\partial_p\bar{q}\rangle$), and the nonlinear effects ($-\langle\omega'\,\partial_p q'\rangle$)) of moisture budget components (mm/day) to precipitation changes averaged over the Zonag Lake and Selin Co basins.

Figures 8e and 8f must be described in more detail for non-expert readers (the explanation of the x-axis comes

in Figure 9, but it should be detailed earlier).

**Response:** Thank you for your suggestion. We have added more detail in caption as "*(e, f) The contribution of each term (precipitation (P'), Evaporation (E'), the change in vertical ($-\langle\omega\partial_p q\rangle'$) and horizontal moisture advection ($-\langle V_h \cdot \nabla_h q\rangle'$), residual ter effects m ($\delta$), thermodynamic effects $-\langle\bar\omega\partial_p q'\rangle$, dynamic ($-\langle\omega'\partial_p\bar q\rangle$), the nonlinear effects ($-\langle\omega'\partial_p q'\rangle$)) of moisture budget components (mm/day) to precipitation changes averaged over the Zonag Lake and Selin Co basins.*"

The legend of Figure 9 is unclear; the letters are not in order, and there is no consistency in the typographical style used (e.g., (i) is not in bold).

**Response:** Thank you. We have revised the legend of Figure 9 (now Figure S4) to ensure that the caption letters are now in the correct order. The updated legend now is: "***Figure S4****. The moisture budget components (mm/day) in 2011. (**a**) precipitation (P'). (**b**) Evaporation (E'). (**c**) residual term. The horizontal moisture advection induced by (**d**) thermodynamic $-\langle\bar\omega\partial_p q'\rangle$, (**e**) dynamic $-\langle\omega'\partial_p\bar q\rangle$ and (**f**) the nonlinear component $-\langle\omega'\partial_p q'\rangle$. The horizontal moisture advection induced by (**g**) moisture changes $-\langle V_h \cdot \nabla_h q'\rangle$ , (**h**) anomalous winds $-\langle V'_h \cdot \nabla_h q\rangle$ and (**i**) the nonlinear component $-\langle V'_h \cdot \nabla_h q'\rangle$*".

---

## Author Response (AR2)

Dear NHESS Editor Olivier Dewitte,

We sincerely appreciate your time and effort in evaluating our manuscript. We are also grateful to the reviewer(s) for their insightful and helpful comments to improve the quality of our manuscript. We have carefully considered the following minor comments from you and the reviewer.

Thank you again for your professional handling of our manuscript.

Yours sincerely,
Prof. Dr. Guoqing Zhang on behalf of all authors

**Editor:**
In addition to the suggestions from the reviewer, I would like to suggest the following technical item for Line 48. Replace "alpine lakes" by "plateau lakes".

**Response**: Done.

**Referee #1**:
(i) exceptional outburst volume 5.42 km^3 - please put this in regional / global context of outburst floods and highlight the magnitude of studied events.

**Response:** We thank the reviewer for this suggestion to provide a visual comparison of the magnitude of this outburst volume. According to a global inventory of glacial lake outburst floods (GLOFs) (Lützow et al. 2023), the largest recorded GLOF occurred in Iceland in 1726, with an estimated volume of ~25 km³, making it the only known event exceeding 5 km³. A similar magnitude is the outburst event of an ice-dammed lake at Russell Fiord, North America, in 1986 with a peak flow of 105,000 m³/s over one hour (Mayo 1989). In the Tibetan Plateau, documented outburst volumes are significantly smaller, with the largest event mainly in the Karakoram, but only ~0.3 km³ (Lützow et al. 2023). The outburst volume of 5.42 km³ from the Zonag event is indeed exceptional in both regional and global contexts. We have added some sentences to emphasis the importance in the Section "4.2 Consequences of Zonag Lake and Selin Co outburst" as follow:

"*The outburst volume of 5.42 km³ from the Zonag event is exceptional in both a regional and a global context. According to a global inventory of glacial lake outburst floods (GLOFs) (Lützow et al. 2023), the largest recorded GLOF occurred in Iceland in 1726, with an estimated volume of ~25 km³, making it the only known event to exceed 5 km³. The outburst of an ice-dammed lake at Russell Fiord, North America, in 1986, with a similar water storage of ~5.4 km³ before breaching, released water with a peak discharge of 105,000 m³/s in one hour (Mayo 1989). In the Tibetan Plateau, documented outburst volumes are much smaller, with the largest event mainly in the Karakoram, but only ~0.3 km³ (Lützow et al. 2023).*".

**References**:

Lützow, N., Veh, G., Korup, O., 2023. A global database of historic glacier lake outburst floods. Earth Syst. Sci. Data, 15(7): 2983-3000. doi:10.5194/essd-15-2983-2023.

Mayo, L.R., 1989. Advance of Hubbard Glacier and 1986 Outburst of Russell Fiord, Alaska, U.S.A. Annals of Glaciology, 13: 189-194. doi:10.3189/S0260305500007874.

(ii) the two new paragraphs in Section 4.3. are confusing. Are the lakes that produced the outbursts really endorheic? Satellite images show that there is surface outflow before the outbursts and the outburst was caused by erosion / breaching of it, no? Because the lakes are having surface outflow, they are not endorheic by definition. If this outflow is seasonal, they are rather perrenial. The whole system is, perhaps, endorheic (?). Please make this clear and revise this rather confusing classification in section 4.3.

**Response**: The lakes producing the outbursts are indeed endorheic by definition, as they have not surface outflow under natural conditions. Although the surface features around these lakes may appear to be an outflow channel on satellite imagery, further zooming out of the image confirms that the lakes are not connected to the channel (see Figure R1 below). The outburst events were caused by erosion and breaching of the natural boundaries of the lake. Furthermore, the region where these lakes are located is part of an endorheic basin. Within this basin, there is a mix of lake types, including both endorheic (dominant in number, with no surface outflow) and exorheic (with clear overflow channels) lakes.

[Figure]

**Figure R1**. Selin Co and pre-existing channels are not connected prior to the outburst.